# StoryBench: A Multifaceted Benchmark for Continuous Story Visualization

**Emanuele Bugliarello**[*,r,ð,c]  **Hernan Moraldo**[ð]  **Ruben Villegas**[ð]

**Mohammad Babaeizadeh**[ð]  **Mohammad Taghi Saffar**[ð]  **Han Zhang**[ð]  **Dumitru Erhan**[ð]

**Vittorio Ferrari**[r]  **Pieter-Jan Kindermans**[ð]  **Paul Voigtlaender**[r]

[r]Google Research  [ð]Google DeepMind  [c]University of Copenhagen

https://github.com/google/storybench

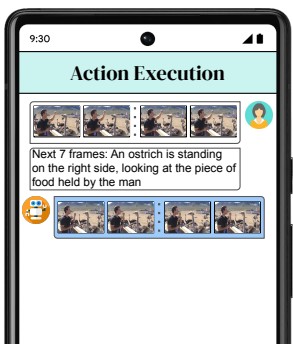
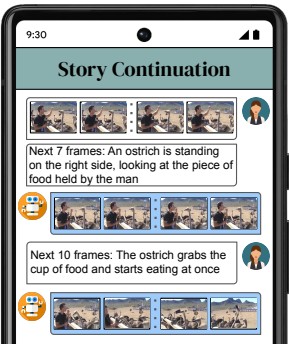
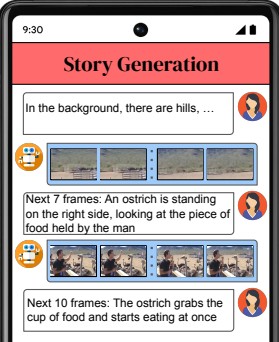

Figure 1: We propose STORYBENCH, a benchmark for continuous story visualization tasks where a model generates a video given a sequence of text prompts and their duration, as well as a video prompt for two tasks.

## Abstract

Generating video stories from text prompts is a complex task. In addition to having high visual quality, videos need to realistically adhere to a sequence of text prompts whilst being consistent throughout the frames. Creating a benchmark for video generation requires data annotated over time, which contrasts with the single caption used often in video datasets. To fill this gap, we collect comprehensive human annotations on three existing datasets, and introduce *StoryBench*: a new, challenging multi-task benchmark to reliably evaluate forthcoming text-to-video models. Our benchmark includes three video generation tasks of increasing difficulty: *action execution*, where the next action must be generated starting from a conditioning video; *story continuation*, where a sequence of actions must be executed starting from a conditioning video; and *story generation*, where a video must be generated from only text prompts. We evaluate small yet strong text-to-video baselines, and show the benefits of training on story-like data algorithmically generated from existing video captions. Finally, we establish guidelines for human evaluation of video stories, and reaffirm the need of better automatic metrics for video generation. StoryBench aims at encouraging future research efforts in this exciting new area.

---

[*]Work completed during an internship at Google. Correspondence to emanuele@di.ku.dk.

37th Conference on Neural Information Processing Systems (NeurIPS 2023) Track on Datasets and Benchmarks.

# 1 Introduction

There has been an explosion in the capabilities of generative AI models in the last two years. Recent language models can produce fluent text [*e.g.*, 1–4], audio generation models can synthesize high-fidelity speech and music [5–7, *i.a.*], and text-to-image models can create photorealistic images [*e.g.*, 8–11], paving the way to real-world applications of AI in art, design and content creation.

Generating high-quality videos of arbitrary duration from text is still, however, an arduous task. Compared to text-to-image synthesis, video generation introduces several distinctive challenges. First, videos should be coherent over time (*e.g.*, objects should be consistent from frame to frame) while reflecting the actions described in text prompts. Second, training text-to-video models is computationally expensive: multiple frames need to be synthesized within the same context to produce smooth transitions over time. And third, video–text datasets [*e.g.*, 12–14] are orders of magnitude smaller than corresponding image–text datasets [15, 16, *i.a.*].

To enable researchers to reliably measure progress in video generation from text prompts that change over time, we introduce the STORYBENCH benchmark. We create STORYBENCH by collecting dense, rich annotations for the validation and test sets of three existing, general video datasets: DiDeMo [17], Oops [18] and UVO [19], for a total of 6K videos. Our human annotators first describe a video with a *sequence* of captions, one for each action, forming the story of the video. In this stage, our annotators also *temporally segment* the full video accordingly, providing timestamps for each caption. This results in unique datasets, with rich annotations of a sequence of actions (*i.e.*, a story) that have clear timestamps. Moreover, we next ask annotators to tag *each* of the resulting 18K video segments according to 34 labels that span 6 categories, such as *camera movements* and *foreground interactions*, allowing researchers and practitioners to easily pinpoint the limitations of their text-to-video models. Finally, given the challenges of obtaining high-quality training data for story visualization, we devise a pipeline to automatically transform existing, grounded video captions [20] into stories, resulting in training data that is very rich in structure, with (i) a sequence of action descriptions, (ii) their timestamps, and even (iii) mouse traces and segmentation masks. We show that this transformation leads to better models, and encourage future work in data curation strategies to improve performance.

Based on these data, we define three tasks of increasing difficulty (see Figure 1) with the goal of synthesizing videos that follow a sequence of text prompts (*i.e.*, a story). In the *action execution* task, a text-to-video model is simply asked to continue a conditioning video from a single caption (*i.e.*, a short sentence describing a single action). This task is extended in *story continuation*, where the model must generate a sequence of coherent video segments given a conditioning video and a series of captions. Finally, the *story generation* task also asks a model to generate a sequence of coherent video segments from a series of captions, but without a conditioning video. Instead, the model is first asked to generate its own conditioning video from a textual description of the expected scene background. For each of the captions, models are also given the expected duration of the corresponding video segment (*e.g.*, 1.5s). This requires models to be *controllable*, resembling real-world applications of artistic content creation. Overall, we name the overarching challenge *continuous story visualization*.

We train a small, 345M parameter text-to-video Phenaki [21] model. In STORYBENCH, models can be evaluated in three different setups: *zero-shot*, where a model is tested after general-purpose pretraining, given no STORYBENCH data; *single-task*, where a model is separately fine-tuned on each dataset; and *multi-task*, where a single model is fine-tuned on our three datasets. Due to the challenges in evaluating videos, we conduct a human study by defining guidelines that capture different desired qualities of continuous story visualization. We also report performance across five automatic metrics, but we show they do not align well with human judgment; reasserting the need of better metrics.

**Contributions.** In this work, we **1)** introduce STORYBENCH, the first benchmark for continuous story visualization, to inspire more work on real-world, text-to-video generation of arbitrary duration through a reproducible and comprehensive setup. **2)** STORYBENCH contains rich human annotations, consisting of action-by-action descriptions, their timestamps, multimodal grounding via mouse traces, and labels for each segment to easily determine failure modes. STORYBENCH is a truly multifaceted benchmark: it spans three tasks of continuous story visualization, across three different datasets, and three evaluation setups (zero-shot, single- and multi-task). We **3)** devise an automatic procedure to transform video captions into a sequence of captions, each describing a single action, and show the effectiveness of using such rich data during model fine-tuning. We **4)** establish a compact yet strong baseline, define guidelines for human evaluation, as well as a set of automatic metrics for reproducible comparisons of the next generation of text-to-video models. **5)** Our results show the

benefit of training text-to-video models to continue videos (rather than generating them from scratch) on story-like data, but they also suggest discrepancies between human and automated evaluation. We invite the community to report results on STORYBENCH at `https://paperswithcode.com/dataset/storybench`. We provide data, annotation instructions, human evaluation guidelines, and code for automatic evaluation at `https://github.com/google/storybench`.

## 2   Related Work

**Text-guided generative models for vision.**   We are currently witnessing tremendous progress in the task of text-to-image generation and editing [8–11, 22–27], fueled by sequence-to-sequence Transformer models [28] and diffusion models [29] trained on massive amounts of image–text data [15]. Research on text-to-video generation has also received attention recently. GODIVA [30] autoregressively generates videos from text using a three-dimensional sparse attention mechanism. NÜWA [31] presents a unified framework for multi-task learning of various generation tasks, including text-to-video. NUWA-Infinity [32] is a generative model that can synthesize arbitrarily-sized images or long-duration videos with an autoregressive over autoregressive generation mechanism. In a similar spirit, NÜWA-XL [33] proposes a diffusion over diffusion approach that allows to generate long videos in parallel through a coarse-to-fine process. CogVideo [34] adds temporal attention modules on top of a frozen text-to-image model to reduce the computational requirements for text-to-video learning. Make-a-Video [35] also starts from a text-to-image model but fine-tunes it while adding pseudo-3D convolution and temporal attention layers. Concurrently, Video Latent Diffusion Models [36] turn pretrained image diffusion models into video generators by fine-tuning them with temporal alignment layers, and Imagen Video [37] generates high definition videos using a cascade of video diffusion models. Ho *et al.* [38] train space-time factorized U-Net models [39] on images and videos, and propose a sampling method to improve longer video generations. Our baselines are based on Phenaki [21], which can generate arbitrary long videos from a sequence of text prompts.

**Story visualization.** In this paper, we propose STORYBENCH, a benchmark for the task of generating a *video* from a sequence of text prompts (*i.e.*, a story), which we refer to as *continuous* story visualization. In the literature, story visualization [40] is the task of generating a sequence of *images* to narrate a multi-sentence story (one image per sentence) with a global visual consistency across dynamic scenes and entities. The authors created two artificial datasets from CLEVR [41] and Pororo [42], and proposed a model based on sequential conditional GANs. To improve story visualization, Maharana *et al.* propose a dual learning framework and a copy mechanism in [43], and leverage grammatical and visual structure as well as commonsense information in [44]. In [45], the authors introduce the DiDeMoSV dataset, and propose to 'retro-fit' a pretrained text-to-image model with task-specific modules to improve on the task of story continuation, resulting in StoryDALL-E. Finally, Rahman *et al.* [46] extend the synthetic MUGEN dataset [47] for multi-sentence storylines, and propose an autoregressive diffusion-based framework with a visual memory module to capture the entities and background context across the generated frames. Unlike previous work, in STORYBENCH, we focus on generating continuous videos (rather than key-frames) on natural (rather than cartoon or synthetic) data. Moreover, we also use DiDeMo to visualize stories but rather than using the existing temporal queries and automatically matching them to key-frames [45], we ask human annotators to thoroughly describe the story of the videos while manually annotating timestamps for each sentence.

## 3   StoryBench

Aiming for a comprehensive resource to assess the ability of generative models to visualize stories, we propose STORYBENCH, the first real-world benchmark for text-to-video story generation. Unlike previous work which frames story visualization as the task of generating a single key-frame per text prompt, STORYBENCH evaluates the ability of generative models to synthesize *continuous*, *natural* videos from a sequence of text prompts. To do so, we collect rich annotations that provide insights and nuances of any model's capabilities, and easily discover failure modes. STORYBENCH consists of three different datasets, three tasks of increasing difficulty, and three evaluation setups.

Generating videos is a very complex task for state-of-the-art models. Some of the key challenges involve generating videos that (i) have a coherent storyline, (ii) are visually realistic, and (iii) can be controlled according to user intent. STORYBENCH aims at benchmarking these three challenges by (i) defining three tasks with increased difficulty in storyline; (ii) focusing on real-world, natural videos; and (iii) enabling control over the generated content through text descriptions and video duration.

| Dataset | #Videos | #Stories (per video) | #Segments (per story) | #Words (per story) | #Labels (per story) | #Actors (per video) |
|---|---|---|---|---|---|---|
| DiDeMo-CSV | 1,399 | 1,399 (1.00) | 4,926 (3.52) | 80,405 (57.47) | 14,463 (10.34) | 3,228 (2.31) |
| Oops-CSV | 1,888 | 3,243 (1.72) | 7,198 (2.22) | 131,485 (40.54) | 30,027 (9.26) | 6,556 (3.47) |
| UVO-CSV | 2,917 | 4,258 (1.46) | 6,227 (1.46) | 122,542 (28.78) | 36,751 (8.63) | 7,743 (2.65) |

Table 1: Statistics of our collected evaluation datasets. Actors refer to entities with a key role a video.

| Category | Labels |
|---|---|
| Camera Movements | static shot, pan, tilt, zoom, tracking shot, aerial shot, point-of-view shot |
| Foreground Entities | people, animals, vehicles, food/drinks, containers, tools |
| Foreground Actions | humans moving, animals moving, objects moving, humans using objects, animals using objects, static actions |
| Background Actions | animate entities moving, objects moving, animate entities using objects, static actions, dynamic background |
| Foreground Interactions | dialogues, direct, indirect, object-based |
| Foreground Transitions | new entities, new objects, entities vanish, objects vanish, entities re-enter, objects re-enter |

Table 2: Overview of categories and labels for each video segment to easily detect failure modes.

## 3.1 Datasets

We annotate three different, real-world, open domain video datasets for continuous story visualization. Table 1 lists high-level statistics of our evaluation data. See App. D for more details and data samples.

**DiDeMo-CSV.** DiDeMo [17] is a dataset collected for the task of temporal localization, where a model is tasked to identify the video segment corresponding to a given text query (*e.g.*, 'the cat jumps on the mat'). The authors define segments with a resolution of 5 seconds. Due to the nature of the temporal localization task, the original text descriptions are often short and lack essential context information to be used for story visualization. We hence ask human annotators to collect descriptions of the actions happening in a video in order to create a coherent story.[2] The annotators are also tasked to (i) add timestamps for each action, (ii) add a description of the background (to be used for the story generation task), and (iii) specify the number of important actors throughout the video.[3]

**Oops-CSV.** Oops [18] is a dataset covering unintentional human actions (*i.e.*, fail videos), originally filmed by nonprofessionals, that are diverse in actions, environments, and intentions. This kind of fail videos are extremely interesting for video generation, as they are unpredictable. A failure is typically a surprising, unexpected action, and one that an ordinary video model would not likely generate. To use these videos for the task of continuous story visualization, we start from the rich annotations in VidLN [20], which consist of descriptive captions that are grounded to the videos through mouse traces. First, we design a pipeline to automatically split the original captions into a sequence of actions that keep most of the original words (see Section 4). Second, we ask human annotators to refine these preprocessed captions by (i) fixing any errors in sentence splitting whilst keeping most of the original words (in order to preserve the rich annotations from VidLN), (ii) adding timestamps for each action, and (iii) adding a concise context description. In fact, the VidLN captions describe the actions of a *single* entity (*i.e.*, actor) throughout the video. Upon inspection, we found that such captions were unsuitable for our task, as they often lacked information that was crucial to synthesize a video similar to the ground-truth. The VidLN data already specified the number of important actors throughout the video, which we update based on stories that were kept after a quality insurance phase.

**UVO-CSV.** UVO [19] is a dataset originally collected for open-world, class-agnostic object segmentation in videos. Videos in UVO are sourced from Kinetics-400 [48], which contains 10-second 30fps videos labeled with human actions. UVO includes many videos with crowded scenes and complex background motions, taken by both professionals and amateurs. As UVO videos have also been annotated with VidLN data, we follow the same steps as for Oops-CSV to create UVO-CSV.

**Segment-level labels for easy diagnostics.** In addition to data for visual generation of stories, we collect rich annotations to promptly detect failure modes of text-to-video models in STORYBENCH. Specifically, together with artists and designers that work with generative AI, we defined 34 labels across 6 categories, shown in Table 2. These include: *camera movements*, to determine the type of shot; *foreground entities*, to know which entities are shown in the video; *foreground* and *background* actions, to know which actions happen in the video; *foreground interactions*, to know how different entities interact with each other; and *foreground transitions*, to know how entities evolve throughout

---

[2]We only annotated videos from the dev and test sets that were still available online as of February 22, 2023.

[3]Important actors are entities (*e.g.*, people, animals) that perform actions central to the story of the video.

the video. For each video segment and label, annotators were asked to check two boxes indicating whether a given label was shown in the video and/or in the caption. By doing so, we can study whether models can capture specific aspects of video generation while also knowing if those aspects were mentioned in the text prompt. We provide further details on each category and label in App. D.1.

**Human annotation framework.** We rely on an in-house platform to annotate our data. For any kind of annotations, human annotators first undergo a training phase. In this phase, they raise any doubts and present corner cases to the first author, who revisits the guidelines with further details and examples. After training, the first author and last author meet with the annotators' managers to ensure the task is clearly understood. The managers then assess which annotators have a high performance for the task, before collecting the final annotations. The annotators' managers, the first author and the last author communicate regularly throughout the annotation process, validating samples and discussing any potential issues (*e.g.*, removing videos with problematic and offensive content).

## 3.2 Tasks

We define three tasks for continuous story visualization of increasing complexity (shown in Figure 1). We call *context* the history of all text and/or video prompts available to a model at any generation step. The *input*, instead, includes a text prompt describing the next video to be generated, and its duration.

**Action execution.** The task of *action execution* simply requires a model to generate the next action specified in the input. The context is given by the ground-truth video preceding the generation step. Whenever the *first* action of a video must be generated, models are conditioned on the first 0.5s of ground-truth video, in order to provide visual context whilst being robust to camera movements.

**Story continuation.** The task of *story continuation* requires a model to generate a video from a *sequence* of inputs and a conditioning video. Analogous to *action execution*, whenever the *first* action of a video must be generated, models are conditioned on the first 0.5s of ground-truth video. The context for the following steps also includes all the previous text prompts and synthesized videos.

**Story generation.** The task of *story generation* also requires a model to generate a video from a *sequence* of inputs, but without any ground-truth visual conditioning. Instead, the context for the first generation is a video synthesized by the model from a text description of the *background*, to guide the space of possible outputs. The following steps are generated analogously to *story continuation* steps.

## 3.3 Human Evaluation

Evaluation of generative models for perceptual data, such as images and videos, is critical to validate their performance. In fact, humans possess complex cognitive abilities to interpret visual content, which automatic metrics often fail to capture. Further, we are ultimately interested in the impact and utility of generative models on users, and human evaluation provides such user-centric perspective.

We follow previous work [10, 21, 22, 35] in doing side-by-side evaluations, where human raters are asked to choose the best (or none) of two outputs for the same prompt. We extend the common metrics of visual quality and text adherence to capture crucial aspects of story visualization as follows.

**Visual quality:** Which video looks better?

**Text adherence:** Which video better reflects the caption?

**Entity consistency:** Throughout which video are entities more consistent (*e.g.*, clothes do not change without a change described in the caption)?

**Background consistency:** In which video is the background more consistent (*e.g.*, the room does not change without a change described in the caption)?

**Action realism:** In which video do actions look more realistic (*e.g.*, according to physics)?

An example of our interface is shown in App. D.3. For each evaluation, humans are provided with the full text story; and each text prompt is rendered as a video below each generated video to facilitate mappings of sentences to their corresponding outputs. For the tasks of *action execution* and *story continuation*, we also provide the conditioning video but add a red border around it to easily identify its end. The models are anonymized and the output pairs are randomly ordered (left vs. right) for each presentation to a rater. We randomly sample 100 stories, and ask three raters to judge each pair.

### 3.4 Automatic Metrics

Human evaluation will always be the gold standard of generative models' evaluation. However, it is both time-consuming and expensive to conduct. To help researchers and practitioners quickly deploy and evaluate new systems, different automatic metrics that correlate fairly well with human evaluations have been proposed over the years, such as FID [49] and FVD [50], as well as CLIP [51] for the recent text-guided models. We consider the following automatic metrics for STORYBENCH.

**Fréchet Image Distance (FID):** The FID score evaluates the quality of generated images (*i.e.*, frames here) by comparing how similar their features are to features of real images. We consider both the standard features given by InceptionV3 [52], as well as features from a stronger CLIP (ViT-L/14).

**Fréchet Video Distance (FVD):** FVD extends the idea behind FID to videos: a generative model must capture the underlying distribution of real-world videos. We consider both the standard features from I3D [53], and features from InternVideo-MM-L-14 [54], a state-of-the-art, video–text model.

**Video–Text Matching (VTM):** We estimate the alignment between generated videos and their text descriptions with the similarity of the visual and textual features given by a multimodal model. We consider both the average cosine similarity of each frame with the corresponding caption given by CLIP (ViT-L/14), and the cosine similarity of a video with its caption given by InternVideo-MM-L-14.

**Perceptual Quality Assessment (PQA):** We also measure *perceptual* quality, which aims at predicting the average human subjective perception of a video. For this, we use DOVER [55], a state-of-the-art model trained to predict the mean opinion score of user-generated videos.

**Similarity of Images (SIM):** For the tasks of *action execution* and *story continuation*, a model is conditioned on a ground-truth video, and asked to continue it for the exact same number of frames as the original one. We can hence directly compare each generated frame with its ground-truth match. We compute frame cosine similarity from the normalized features extracted to compute FID.

We only compute automatic metrics on the generated videos, disregarding the conditioning ground-truth video (*action execution* and *story continuation* tasks). For *story generation*, the frames generated from the 'background' descriptions are omitted, and we do not report SIM as it is ill-defined here. We release our evaluation code to promote reproducible and comparable research on STORYBENCH.

## 4 Training Data Challenge

In addition to the challenges of video generation discussed in Section 3, the lack of large amounts of high-quality data is a major bottleneck to train strong text-to-video models. Manually collecting annotations for videos is both expensive and time-consuming, especially in story-like format.

Given the major role that data plays in training state-of-the-art generative models, and the impact that careful pre-processing and synthetic data can have, we explicitly define the challenge of training data curation to improve performance in STORYBENCH.

We provide a first approach in this work, which we will show to be beneficial in Section 6. Specifically, we define an automatic pipeline to transform the original VidLN captions for Oops and UVO into multiple sentences, each approximately describing a single action, and determine their timestamps.

To create an original VidLN annotation [20], the annotator first watched the video and selected important actors (*e.g.*, man or dog). For each actor they would then select a few key-frames that are representative of the actions performed by this actor throughout the video. They then described the actions of that entity with their voice, while at the same time moving their mouse over the key-frames to provide a spatio-temporal localization for each word. Finally, they transcribed their audio for a high-quality caption. For each actor, VidLN provides a long caption that often describes multiple actions. For the purpose of story visualization, we want to split the long caption into individual actions. Furthermore, we need timestamps for each action. The VidLN captions do not provide explicit timestamps for actions, but several words are localized in space–time on one or multiple key-frames. We use this indirect temporal information to derive an approximate timestamp.

Our pipeline consists of five steps (see Figure 2). **1)** We prompt Flan-U-PaLM 540B [3, 56] – an instruction-tuned large language model (LLM) – to split a given description into multiple sentences, each (approximately) describing a single action, whilst making minimal (ideally, no) changes to the original words. **2)** We can then map every word in each sentence to all (if any) of its associated

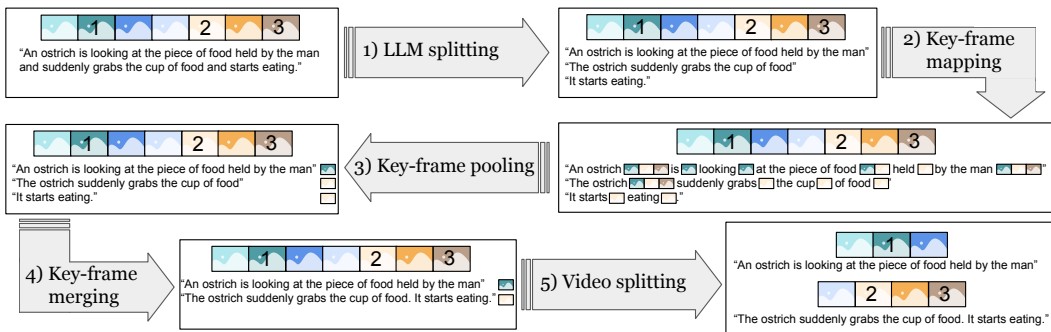

Figure 2: Automatic pipeline to transform VidLN annotations into stories. Starting from a video, its caption and annotated key-frames, we use an LLM to split the caption into multiple sentences. We then transfer the key-frames of the original caption into the new ones. We select a single key-frame per caption, merge captions with the same key-frame and finally split the video accordingly.

key-frames. As the LLM sometimes modifies words for better grammaticality, we rely on lemmas instead. **3)** We associate a single key-frame to each caption, determined by the frame most commonly referenced by the verbs (if available), or across all words in the sentence. **4)** If a continuous sub-sequence of sentences maps to the same key-frame, we concatenate those sentences into a single one. **5)** Finally, we map each sentence to a time interval using the timestamp of its associated key-frame.

Our code to transform VidLN descriptions into stories is online. See App. D.5 for robustness statistics.

## 5 Experimental Setup

We train a compact (345M parameters) yet strong Phenaki [21] baseline for STORYBENCH, which we refer to as PHENAKI-GEN. It consists of a C-ViViT encoder, a text encoder, and a MaskGiT model.

The goal of the C-ViViT encoder is to learn a codebook of visual features to compress the videos in both temporal and spatial dimensions, while staying autoregressive in time. We train a 52M C-ViViT model on 27M Web-scraped video samples. Rather than a square resolution [21], we opt for a more natural, rectangular resolution of $160 \times 96$ pixels. After being trained, the C-ViViT is kept frozen.

MaskGiT [57] is a bidirectional Transformer trained to predict multiple masked video tokens simultaneously conditioned on the caption features computed by the text encoder. Compared to the MaskGiT model used in the original Phenaki model (1.8B), ours is much smaller at 345M (24 layers). In contrast to the original Phenaki, which used a pretrained text encoder, we train the text encoder jointly with the video generation model on our collection of Web-collected 27M video captions. Our text encoder is a 12-layer Transformer that uses a T5 tokenizer [58]. Both models use 16 attention heads, MLP size of 3072, and default embedding size of 768. The maximum sequence length for the video component is 1440 tokens (11 compressed frames), and 64 tokens for the text component. The model is trained for 3M steps with a batch size of 256 using a fixed learning rate of 4.5e-5 after a linear warm-up for the first 10K steps using the Adam [59] optimizer. After pretraining, PHENAKI-GEN can generate arbitrarily long videos using the last 5 frames to condition the generation of the next 6.

We fine-tune PHENAKI-GEN on the training data of STORYBENCH for 500K steps with a much smaller batch size of 64 samples, using the same learning rate and optimizer of the pretraining stage.

Furthermore, we explore a novel variant of our model that is fine-tuned for *continuation*, PHENAKI-CONT, using the same hyperparameters as PHENAKI-GEN. Rather than training to predict all the input tokens, we only mask and generate the tokens of the last 6 frames, conditioning on 5 ground-truth frames. We hypothesize that this training would result in better transitions and consistency.

## 6 Results

In this section, we discuss the performance of our baselines on STORYBENCH evaluated by humans and automatically. We append -ZS for results obtained in the *zero-shot* setting, -ST for *single-task* fine-tuning, and -MT for *multi-task* fine-tuning. In particular, we start by discussing performance on Oops-CSV (shown in Table 4) due to space limitations. We then also comment on the overall

PROMPT: The llama is putting his mouth in a white bowl in the car.

PHENAKI-GEN-ZS

PHENAKI-CONT-ST

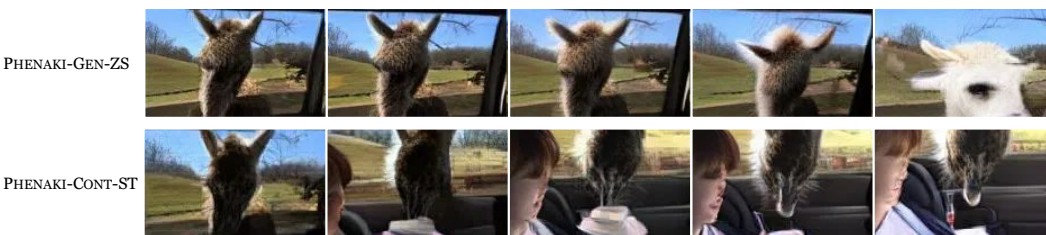

Figure 3: Comparison of PHENAKI-GEN-ZS to PHENAKI-CONT-ST on a prompt for *action execution*. While PHENAKI-GEN-ZS animates the animal, it adheres less to the text prompt, and the llama changes over time, PHENAKI-CONT-ST successfully shows two entities from the context (person and bowl), while keeping the animal and the surroundings consistent. Video subsampled by a factor 4.

PROMPTS:
A girl wearing a green-black bikini is walking on a diving board others are swimming
The girl slips on it and others get scared
The girl falls in the sea while the other man is wearing glass and swimming

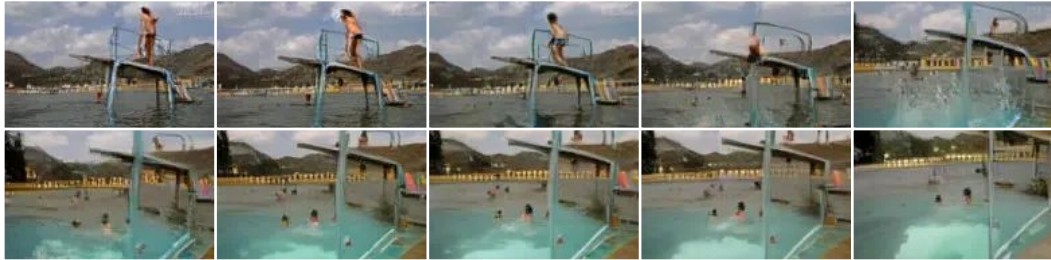

Figure 4: Applying PHENAKI-CONT-ST to a longer sequence with multiple prompts for story continuation. The model can generate the right action, including the water splash from the girl falling in the water. The background is kept relatively consistent within a short time frame, but it starts changing at longer time scales. The video was subsampled by a factor 4 to be shown here.

findings, reported in Figure 17 (App. E.1) for human evaluation, Tables 5 to 7 when using a set of neural metrics, and in Tables 8 to 10 (App. E.2) when using the others. Moreover, we also benchmark a variant of PHENAKI-CONT fine-tuned on the original Oops dataset, rather than after our automatic story-like transformation, which we refer to as PHENAKI-CONT-ST-orig. For every story, each model generates 4 output videos. We randomly sample one of them for human evaluation, but report mean and standard deviation across all 4 generated videos for automatic metrics. Due to our setup, all the results are at 8fps and using a $160 \times 96$ pixel resolution for the generated videos, but note that ground-truth videos were processed at their original resolution for automatic evaluation.[4]

## 6.1 Human Evaluation

Table 3 shows our human evaluation (*c.f*. Section 3.3) results for story continuation on Oops-CSV. Our model fine-tuned in continuation mode is vastly preferred over the zero-shot PHENAKI-GEN-ZS in all metrics. Figure 3 illustrates an example of why raters prefer PHENAKI-CONT-ST over PHENAKI-GEN-ZS. The improvement in long-term text adherence and consistency, however, has its limitations. We illustrate this in Figure 4: while the key actions are executed correctly, the background tends to drift over time, showing that our modest sized model has still large margins for improvement. We also see that human raters consistently prefer PHENAKI-CONT-ST over PHENAKI-GEN-ST (both fine-tuned on the same data), proving the effectiveness of the novel fine-tuning in continuation mode.

However, fine-tuning on all datasets jointly (PHENAKI-CONT-MT) results in videos of comparable quality as the zero-shot model. This is a common limitation of multi-task systems, and both a larger model and longer training could lead to better performance. Finally, we compare PHENAKI-

---

[4]For instance, the most common resolutions are $1280 \times 720$ for Oops-CSV Dev set, $854 \times 480$ for UVO-CSV Dev set, and $360 \times 640$ for DiDeMo-CSV Dev set. We release our extracted features of the ground-truth videos for all the automatic evaluation metrics online at `https://github.com/google/storybench`.

| Human Eval \| Oops-CSV \| Story Continuation | Visual quality | | | Text adherence | | | Entities consistency | | | BG consistency | | | Actions realism | | |
| L vs R | L | Tie | R | L | Tie | R | L | Tie | R | L | Tie | R | L | Tie | R |
|---|---|---|---|---|---|---|---|---|---|---|---|---|---|---|---|
| PHENAKI-GEN-ZS vs PHENAKI-CONT-ST | 37 | 14 | 49 | 34 | 8 | 58 | 29 | 15 | 56 | 16 | 19 | 65 | 29 | 13 | 58 |
| PHENAKI-GEN-ST vs PHENAKI-CONT-ST | 25 | 4 | 71 | 21 | 3 | 76 | 19 | 18 | 63 | 7 | 28 | 65 | 20 | 8 | 72 |
| PHENAKI-GEN-ZS vs PHENAKI-CONT-MT | 51 | 4 | 45 | 46 | 8 | 46 | 33 | 23 | 44 | 24 | 36 | 40 | 45 | 7 | 48 |
| PHENAKI-GEN-ZS vs PHENAKI-CONT-ST-orig | 53 | 4 | 43 | 41 | 10 | 49 | 33 | 28 | 39 | 29 | 35 | 36 | 38 | 7 | 55 |

Table 3: Human evaluation results for story continuation on Oops-CSV. For 100 randomly selected stories, we show the generated videos to 3 raters and report their majority vote. For each compared pair of models, L refers to the left model and R to the right model.

| Oops-CSV | Action Execution | | | | | Story Continuation | | | | | Story Generation | | | | |
| Model (@8 fps) | FID↓ | FVD↓ | SIM↑ | PQA↑ | VTM↑ | FID↓ | FVD↓ | SIM↑ | PQA↑ | VTM↑ | FID↓ | FVD↓ | SIM↑ | PQA↑ | VTM↑ |
|---|---|---|---|---|---|---|---|---|---|---|---|---|---|---|---|
| *Zero-shot* | | | | | | | | | | | | | | | |
| PHENAKI-GEN-ZS | 167 | 416 | 72.8 | **5.8** | **22.1** | 277 | 623 | 70.3 | **7.2** | **21.7** | 310 | 933 | N/A | **8.1** | 21.0 |
| *Single-Task (fine-tuned on Oops-CSV)* | | | | | | | | | | | | | | | |
| PHENAKI-GEN-ST | 177 | 446 | 72.3 | 4.0 | 21.5 | 250 | 589 | 70.0 | 4.3 | 21.3 | 246 | **614** | N/A | 4.3 | **21.1** |
| PHENAKI-CONT-ST | **114** | **350** | **73.2** | 4.9 | 21.5 | **155** | **488** | **71.1** | 5.3 | 21.2 | **171** | 711 | N/A | 5.4 | 19.4 |
| *Multi-Task (fine-tuned jointly on Oops-CSV, DiDeMo-CSV, and UVO-CSV)* | | | | | | | | | | | | | | | |
| PHENAKI-CONT-MT | 140 | 353 | 72.8 | 4.7 | 21.7 | 198 | 511 | 70.6 | 5.1 | 21.4 | 201 | 860 | N/A | 5.0 | 19.4 |

Table 4: Results from automatic evaluation metrics on Oops-CSV tasks. Best results are in **bold**. FID and SIM use InceptionV3, FVD uses I3D, PQA uses DOVER, and VTM uses CLIP.

CONT-ST-orig, fine-tuned on the original Oops VidLN annotations, against the zero-shot model. PHENAKI-CONT-ST-orig still outperforms the zero-shot model in all criteria except for visual quality. Compared to PHENAKI-CONT-ST, however, its margins over the zero-shot model are much smaller, clearly demonstrating the benefit of our training data transformation pipeline (*c.f.* Section 4).

In App. E.1, we report human ratings between PHENAKI-GEN-ZS and PHENAKI-CONT-ST on action recognition and story generation. For action recognition, we find consistent results with story continuation. In contrast, the zero-shot model is generally preferred for story generation. We hypothesize that fine-tuning for continuation leads to a less capable model for the task of generation from scratch. Unifying both approaches in a single model is an interesting direction for future work.

## 6.2 Automatic Evaluation

While the results from human raters were very clear, they are expensive to obtain. We hence also report typical automatic metrics on Oops-CSV in Table 4, and on all datasets in Tables 5 to 7.[5]

In Table 4, our model fine-tuned on Oops-CSV in continuation mode (PHENAKI-CONT-ST) performs the best on action execution and story continuation according to FID, FVD, and SIM, again demonstrating the effectiveness of the novel continuation mode. On the other hand, for story generation, the continuation mode yields an improvement in FID but not in FVD. This contrasts with the human raters, who preferred PHENAKI-GEN-ZS in all metrics for story generation (see Figure 17 in App. E.1).

Across all datasets and tasks, PHENAKI-CONT-MT achieves overall strong results, showcasing the effectiveness of training a single text-to-video model in continuation mode on all datasets. On the other hand, PHENAKI-GEN-ZS achieves higher PQA scores, while human raters typically found all the models to have similar *visual quality*. This, however, is partially reflected in our SIM score, which only varies by a small amount between different models (all within 1pp). Nevertheless, while human raters vastly preferred PHENAKI-CONT-ST over PHENAKI-GEN-ZS, the SIM, PQA and VTM metrics do not reflect this, showing low correlation of automatic metrics with human judgments.

## 7 Limitations

STORYBENCH is the first effort towards a multifaceted benchmark for text-to-video generation. It aims at fostering progress on real-world video generation, with dynamic text prompts and controllable duration provided by the users. Yet, it has a few notable limitations. First, while our benchmark tests for multi-sentence generation, the annotated videos are not very long. Second, they are mostly single-shot (*i.e.*, without changes of scene) and user-generated (rather than professional), which

---

[5]We find similar results with newer neural metrics for automatic evaluation in Tables 8 to 10 in App. E.2.

**Action Execution**

| Model (@8 fps) | Oops-CSV | | | | | UVO-CSV | | | | | DiDeMo-CSV | | | | |
|---|---|---|---|---|---|---|---|---|---|---|---|---|---|---|---|
| | $\text{FID}_{Iv3}\downarrow$ | $\text{FVD}_{I3D}\downarrow$ | $\text{SIM}_{Iv3}\uparrow$ | $\text{PQA}\uparrow$ | $\text{VTM}_C\uparrow$ | $\text{FID}_{Iv3}\downarrow$ | $\text{FVD}_{I3D}\downarrow$ | $\text{SIM}_{Iv3}\uparrow$ | $\text{PQA}\uparrow$ | $\text{VTM}_C\uparrow$ | $\text{FID}_{Iv3}\downarrow$ | $\text{FVD}_{I3D}\downarrow$ | $\text{SIM}_{Iv3}\uparrow$ | $\text{PQA}\uparrow$ | $\text{VTM}_C\uparrow$ |
| *Zero-Shot* | | | | | | | | | | | | | | | |
| PHENAKI-GEN-ZS | $167_{\pm3.53}$ | $416_{\pm8.78}$ | $72.8_{\pm0.04}$ | $\mathbf{5.8}_{\pm0.03}$ | $22.1_{\pm0.05}$ | $110_{\pm0.97}$ | $472_{\pm8.03}$ | $\mathbf{73.0}_{\pm0.02}$ | $\mathbf{8.5}_{\pm0.1}$ | $\mathbf{22.4}_{\pm0.03}$ | $195_{\pm1.57}$ | $454_{\pm3.21}$ | $70.7_{\pm0.03}$ | $\mathbf{6.7}_{\pm0.02}$ | $22.1_{\pm0.02}$ |
| *Single-Task* | | | | | | | | | | | | | | | |
| PHENAKI-GEN-ST | $177_{\pm0.60}$ | $446_{\pm3.47}$ | $72.3_{\pm0.01}$ | $4.0_{\pm0.02}$ | $21.5_{\pm0.00}$ | $165_{\pm0.68}$ | $1004_{\pm8.55}$ | $72.0_{\pm0.01}$ | $4.8_{\pm0.05}$ | $20.1_{\pm0.02}$ | $159_{\pm1.73}$ | $510_{\pm7.09}$ | $70.8_{\pm0.02}$ | $4.5_{\pm0.02}$ | $\mathbf{22.3}_{\pm0.01}$ |
| PHENAKI-CONT-ST | $\mathbf{114}_{\pm1.10}$ | $\mathbf{350}_{\pm2.93}$ | $\mathbf{73.2}_{\pm0.01}$ | $4.8_{\pm0.02}$ | $21.5_{\pm0.02}$ | $178_{\pm1.74}$ | $828_{\pm5.98}$ | $72.0_{\pm0.01}$ | $5.6_{\pm0.02}$ | $20.5_{\pm0.04}$ | $\mathbf{136}_{\pm0.80}$ | $\mathbf{365}_{\pm2.56}$ | $\mathbf{71.2}_{\pm0.01}$ | $5.6_{\pm0.01}$ | $22.1_{\pm0.02}$ |
| *Multi-Task* | | | | | | | | | | | | | | | |
| PHENAKI-GEN-MT | $270_{\pm3.11}$ | $502_{\pm4.65}$ | $71.5_{\pm0.02}$ | $3.8_{\pm0.04}$ | $21.4_{\pm0.03}$ | $235_{\pm2.09}$ | $916_{\pm4.47}$ | $71.4_{\pm0.02}$ | $5.1_{\pm0.05}$ | $20.6_{\pm0.02}$ | $219_{\pm1.65}$ | $510_{\pm1.86}$ | $70.2_{\pm0.02}$ | $4.7_{\pm0.06}$ | $21.9_{\pm0.02}$ |
| PHENAKI-CONT-MT | $140_{\pm1.21}$ | $353_{\pm2.83}$ | $72.8_{\pm0.02}$ | $4.7_{\pm0.01}$ | $21.7_{\pm0.03}$ | $111_{\pm2.01}$ | $\mathbf{435}_{\pm4.89}$ | $72.8_{\pm0.03}$ | $6.8_{\pm0.01}$ | $21.6_{\pm0.03}$ | $166_{\pm3.00}$ | $381_{\pm2.96}$ | $70.9_{\pm0.02}$ | $5.8_{\pm0.03}$ | $21.9_{\pm0.02}$ |

Table 5: Results from automatic evaluation metrics on *action execution* tasks. Best results are in **bold**. FID and SIM use InceptionV3, FVD uses I3D, PQA uses DOVER, and VTM uses CLIP.

**Story Continuation**

| Model (@8 fps) | Oops-CSV | | | | | UVO-CSV | | | | | DiDeMo-CSV | | | | |
|---|---|---|---|---|---|---|---|---|---|---|---|---|---|---|---|
| | $\text{FID}_{Iv3}\downarrow$ | $\text{FVD}_{I3D}\downarrow$ | $\text{SIM}_{Iv3}\uparrow$ | $\text{PQA}\uparrow$ | $\text{VTM}_C\uparrow$ | $\text{FID}_{Iv3}\downarrow$ | $\text{FVD}_{I3D}\downarrow$ | $\text{SIM}_{Iv3}\uparrow$ | $\text{PQA}\uparrow$ | $\text{VTM}_C\uparrow$ | $\text{FID}_{Iv3}\downarrow$ | $\text{FVD}_{I3D}\downarrow$ | $\text{SIM}_{Iv3}\uparrow$ | $\text{PQA}\uparrow$ | $\text{VTM}_C\uparrow$ |
| *Zero-Shot* | | | | | | | | | | | | | | | |
| PHENAKI-GEN-ZS | $277_{\pm4.04}$ | $623_{\pm9.93}$ | $70.3_{\pm0.02}$ | $\mathbf{7.2}_{\pm0.06}$ | $21.7_{\pm0.02}$ | $138_{\pm1.79}$ | $660_{\pm8.20}$ | $\mathbf{72.0}_{\pm0.02}$ | $\mathbf{9.4}_{\pm0.09}$ | $\mathbf{22.2}_{\pm0.04}$ | $341_{\pm1.89}$ | $817_{\pm7.76}$ | $68.0_{\pm0.01}$ | $\mathbf{7.3}_{\pm0.07}$ | $21.6_{\pm0.02}$ |
| *Single-Task* | | | | | | | | | | | | | | | |
| PHENAKI-GEN-ST | $250_{\pm2.05}$ | $589_{\pm6.08}$ | $70.0_{\pm0.02}$ | $4.3_{\pm0.02}$ | $21.3_{\pm0.01}$ | $193_{\pm1.20}$ | $1267_{\pm9.36}$ | $71.0_{\pm0.01}$ | $5.0_{\pm0.03}$ | $19.9_{\pm0.01}$ | $238_{\pm1.24}$ | $690_{\pm1.90}$ | $68.2_{\pm0.02}$ | $4.0_{\pm0.02}$ | $\mathbf{22.0}_{\pm0.01}$ |
| PHENAKI-CONT-ST | $\mathbf{155}_{\pm1.07}$ | $\mathbf{488}_{\pm2.12}$ | $\mathbf{71.1}_{\pm0.03}$ | $5.3_{\pm0.04}$ | $21.2_{\pm0.04}$ | $197_{\pm1.22}$ | $1125_{\pm9.84}$ | $71.1_{\pm0.02}$ | $5.7_{\pm0.03}$ | $20.2_{\pm0.03}$ | $\mathbf{206}_{\pm1.95}$ | $\mathbf{580}_{\pm4.30}$ | $\mathbf{68.7}_{\pm0.01}$ | $5.4_{\pm0.03}$ | $21.6_{\pm0.04}$ |
| *Multi-Task* | | | | | | | | | | | | | | | |
| PHENAKI-GEN-MT | $364_{\pm2.37}$ | $708_{\pm7.29}$ | $69.0_{\pm0.01}$ | $4.2_{\pm0.02}$ | $21.1_{\pm0.02}$ | $268_{\pm2.76}$ | $1188_{\pm8.76}$ | $70.3_{\pm0.02}$ | $5.2_{\pm0.03}$ | $20.5_{\pm0.02}$ | $330_{\pm2.09}$ | $717_{\pm9.12}$ | $67.5_{\pm0.01}$ | $4.3_{\pm0.04}$ | $21.5_{\pm0.02}$ |
| PHENAKI-CONT-MT | $198_{\pm1.97}$ | $511_{\pm5.46}$ | $70.6_{\pm0.04}$ | $5.1_{\pm0.02}$ | $21.4_{\pm0.02}$ | $\mathbf{124}_{\pm2.56}$ | $\mathbf{618}_{\pm6.59}$ | $71.9_{\pm0.03}$ | $7.0_{\pm0.02}$ | $21.3_{\pm0.03}$ | $256_{\pm3.27}$ | $684_{\pm3.15}$ | $68.3_{\pm0.03}$ | $5.5_{\pm0.09}$ | $21.2_{\pm0.01}$ |

Table 6: Results from automatic evaluation metrics on *story continuation* tasks. Best results are in **bold**. FID and SIM use InceptionV3, FVD uses I3D, PQA uses DOVER, and VTM uses CLIP.

**Story Generation**

| Model (@8 fps) | Oops-CSV | | | | | UVO-CSV | | | | | DiDeMo-CSV | | | | |
|---|---|---|---|---|---|---|---|---|---|---|---|---|---|---|---|
| | $\text{FID}_{Iv3}\downarrow$ | $\text{FVD}_{I3D}\downarrow$ | $\text{SIM}_{Iv3}\uparrow$ | $\text{PQA}\uparrow$ | $\text{VTM}_C\uparrow$ | $\text{FID}_{Iv3}\downarrow$ | $\text{FVD}_{I3D}\downarrow$ | $\text{SIM}_{Iv3}\uparrow$ | $\text{PQA}\uparrow$ | $\text{VTM}_C\uparrow$ | $\text{FID}_{Iv3}\downarrow$ | $\text{FVD}_{I3D}\downarrow$ | $\text{SIM}_{Iv3}\uparrow$ | $\text{PQA}\uparrow$ | $\text{VTM}_C\uparrow$ |
| *Zero-Shot* | | | | | | | | | | | | | | | |
| PHENAKI-GEN-ZS | $310_{\pm3.13}$ | $933_{\pm4.50}$ | N/A | $\mathbf{8.1}_{\pm0.03}$ | $21.0_{\pm0.05}$ | $181_{\pm3.87}$ | $1118_{\pm6.13}$ | N/A | $\mathbf{10.0}_{\pm0.06}$ | $\mathbf{20.8}_{\pm0.09}$ | $357_{\pm2.79}$ | $930_{\pm8.71}$ | N/A | $\mathbf{7.6}_{\pm0.08}$ | $21.5_{\pm0.04}$ |
| *Single-Task* | | | | | | | | | | | | | | | |
| PHENAKI-GEN-ST | $246_{\pm1.07}$ | $\mathbf{614}_{\pm6.86}$ | N/A | $4.2_{\pm0.02}$ | $\mathbf{21.1}_{\pm0.03}$ | $188_{\pm1.45}$ | $1371_{\pm6.06}$ | N/A | $4.9_{\pm0.01}$ | $19.8_{\pm0.01}$ | $241_{\pm0.90}$ | $710_{\pm3.54}$ | N/A | $4.0_{\pm0.01}$ | $\mathbf{22.1}_{\pm0.01}$ |
| PHENAKI-CONT-ST | $\mathbf{171}_{\pm5.32}$ | $711_{\pm9.55}$ | N/A | $5.4_{\pm0.03}$ | $19.4_{\pm0.02}$ | $207_{\pm2.75}$ | $1600_{\pm9.97}$ | N/A | $5.4_{\pm0.02}$ | $19.1_{\pm0.04}$ | $\mathbf{232}_{\pm3.11}$ | $\mathbf{689}_{\pm5.69}$ | N/A | $5.4_{\pm0.05}$ | $21.1_{\pm0.04}$ |
| *Multi-Task* | | | | | | | | | | | | | | | |
| PHENAKI-GEN-MT | $356_{\pm4.36}$ | $760_{\pm4.60}$ | N/A | $4.1_{\pm0.02}$ | $\mathbf{21.1}_{\pm0.04}$ | $268_{\pm2.50}$ | $1311_{\pm7.41}$ | N/A | $5.1_{\pm0.03}$ | $20.3_{\pm0.02}$ | $334_{\pm1.16}$ | $739_{\pm4.49}$ | N/A | $4.3_{\pm0.06}$ | $21.6_{\pm0.01}$ |
| PHENAKI-CONT-MT | $201_{\pm5.56}$ | $860_{\pm9.86}$ | N/A | $5.0_{\pm0.11}$ | $19.4_{\pm0.05}$ | $\mathbf{148}_{\pm5.50}$ | $1150_{\pm9.98}$ | N/A | $6.3_{\pm0.08}$ | $18.8_{\pm0.06}$ | $269_{\pm2.80}$ | $863_{\pm9.94}$ | N/A | $5.2_{\pm0.03}$ | $20.4_{\pm0.03}$ |

Table 7: Results from automatic evaluation metrics on *story generation* tasks. Best results are in **bold**. FID and SIM use InceptionV3, FVD uses I3D, PQA uses DOVER, and VTM uses CLIP.

differ from realistic filmmaking applications. Moreover, we highlight that STORYBENCH focuses on real-world videos, and does not include artistic content never seen in the real world. Yet, we expect it can drive general progress in text-to-video generation thanks to its complexity. Our annotation framework would not scale to long videos, and we encourage future work to investigate more efficient protocols. We also note that our baseline models are small (345M) compared to current text-to-image models (20B). Effectively training large text-to-video models is an open question. Finally, we also found that automatic metrics do not fully align with human ratings. This might indicate that the metrics are suboptimal, but we cannot exclude that they would perform better on higher resolution videos. This highlights the need for a future in-depth study on human versus automated evaluation.

# 8   Conclusion

In this work, we collected annotations for the task of video generation from a sequence of text prompts narrating the video evolution (*i.e.*, continuous story visualization). Our data is very rich, with timestamps for each text prompt, as well as diagnostic labels for each video segment. On top of such unique data, we propose STORYBENCH: a new benchmark to measure progress of forthcoming text-to-video models on different tasks, three datasets and in three evaluation setups. STORYBENCH allows to evaluate key challenges of arbitrarily long video generation, such as consistency over time and realistic action synthesis. We benchmarked small yet strong baselines, and showed that fine-tuning a Phenaki model for continuation improves such aspects. Our results highlight a discrepancy between human and automatic ratings. While human evaluation remains the most reliable method, we encourage future work to also report automatic metrics in an effort to better understand and develop effective automatic evaluation. Our benchmark enables a systematic comparison of text-to-video models that can perform story generation with prompts that vary over time and specified duration. Beyond modeling, opportunities for future work include expanding the benchmark with datasets of longer videos and richer variety (*e.g.*, camera jumps, scene changes, etc.) to enhance its applicability to industrial-level videos, and devising efficient data annotations protocols to facilitate the collection.

## Acknowledgments and Disclosure of Funding

We would like to thank Irina Blok, Alonso Martinez, Mario Lučić, Cordelia Schmid, Jasper Uijlings, Andreas Steiner and Lucas Beyer for helpful discussions, as well as the annotators for their work.

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

# A  Author Statement

We bear all responsibilities for the content, licensing, distribution, and maintenance of our datasets in STORYBENCH. Our datasets are released under a CC-BY-4.0 license, and our code under an Apache license. Data, code and annotation guidelines are hosted on GitHub at the following URL: `https://github.com/google/storybench`.

# B  Ethics Statement

The aim of STORYBENCH is to enable reliable measurements of progress in generative text-to-video models. While this kind of models have great potential to assist and augment human creativity [60], there are broader societal issues that need to be considered when developing these models.

First, while we annotate an evaluation set, training current, strong text-to-video models is computationally expensive. This affects not only their financial cost (*e.g.*, hardware and electricity), but also their environmental cost due to the carbon footprint of modern tensor processing hardware [61].

Second, massive amounts of data are required to train state-of-the-art generative models. Such datasets are harvested from the Web, which tend to reflect social stereotypes, oppressive viewpoints, and harmful associations to marginalized identity groups [62–64]. Other biases include those introduced by the use of examples that primarily have English texts and may reflect North American and Western European cultures [65]. We expect models trained on them to reflect these biases, and hence caution developers to assess the limitations of their models before integrating them into user-facing applications. To facilitate positive and safe integration of text-to-video models, we encourage future work to create benchmark evaluations to assess social and cultural biases of these technologies.

While multimodal models can unlock creative applications that can benefit humanity, they can also enable harmful applications. These include surveillance, especially when people are recorded and the recordings are used without their consent, or generation of harmful content, such as pornographic material. A particularly sensitive topic in this space is disinformation. When model outputs achieve realistic quality, they can be used to create convincing fake content (*i.e.*, deepfakes). These can be exploited to spread fake news, defame individuals or portray false situations. To mitigate these harms, watermarks can be applied to every generated video [66] such that it is possible to to identify whether any given video is generated by a particular model.

Due to the impacts and limitations described above, we remark that STORYBENCH aims to measure progress in text-to-video research. For the same reasons, we do not release our baselines to the public. By no means should our data be extended for use in sensitive domains, but rather for creative goals. We believe that generative technologies like the type of text-to-video models that can be evaluated in STORYBENCH can become useful tools to enhance human productivity and creativity.

The collection of our datasets has been enabled by the careful work of several participants. Due to privacy concerns, we did not include the estimated hourly wage paid to them or the total amount spent on participant compensation. We feel that individuals' hourly wage or compensation is personal information and we cannot disclose this under privacy law. However, this work was carried out by paid contractors, and we can confirm that they received their standard contracted wage, which is above the living wage in their country of employment.

# C  Datasheet

**Motivation**

**Q1 For what purpose was the dataset created?** Was there a specific task in mind? Was there a specific gap that needed to be filled? Please provide a description.

STORYBENCH was created to encourage reproducible progress in text-to-video modeling. Existing video captioning datasets consist of a single sentence describing the salient events that happen throughout the entire video. Existing dense video captioning datasets, instead, are either domain-specific (*e.g.*, instructional video) or contain captions that lack enough information to generate a video. With our annotation protocol, we describe each action separately and also map it to a precise timestamp interval, allowing us to evaluate the ability of text-to-video models to generate arbitrarily long stories. Our task of continuous story visualization is closely related to the existing one of story

visualization, which was, however, limited to generate a single key-frame per caption, rather than a continuous video. With the release of STORYBENCH, we aim to establish a framework for reliable evaluation of forthcoming generative video technologies.

**Q2 Who created the dataset (e.g., which team, research group) and on behalf of which entity (e.g., company, institution, organization)?**

STORYBENCH was collected by Google Research.

**Q3 Who funded the creation of the dataset?** *If there is an associated grant, please provide the name of the grantor and the grant name and number.*

Google Research funded the creation of STORYBENCH.

**Q4 Any other comments?**

No.

**Composition**

**Q5 What do the instances that comprise the dataset represent (e.g., documents, photos, people, countries)?** *Are there multiple types of instances (e.g., movies, users, and ratings; people and interactions between them; nodes and edges)? Please provide a description.*

We provide 8,900 annotations of stories across six splits. Each story contains an object serialized in JSON with the following fields: `sentence_parts`, `start_times`, `end_times`, `original_combined_sentence`, `clip_start_time`, `clip_end_time`, `story_number`, `background_description`, `dataset_name`, `video_name`, `vidln_id`, `question_info`, `num_actors_in_video`, `segment_categories`. We provide a description of each field in the README file of our code online. In addition to our annotations, an instance of the dataset requires the corresponding video file from existing datasets.

**Q6 How many instances are there in total (of each type, if appropriate)?**

The DiDeMo-CSV dev split has 744/744 videos/stories, and the test split has 655/655 videos/stories. The Oops-CSV dev split has 979/1578 videos/stories, and the test split has 979/1578 videos/stories. The UVO-CSV dev split has 1019/1665 videos/stories, and the test split has 1565/2613 videos/stories.

**Q7 Does the dataset contain all possible instances or is it a sample (not necessarily random) of instances from a larger set?** *If the dataset is a sample, then what is the larger set? Is the sample representative of the larger set (e.g., geographic coverage)? If so, please describe how this representativeness was validated/verified. If it is not representative of the larger set, please describe why not (e.g., to cover a more diverse range of instances, because instances were withheld or unavailable).*

STORYBENCH consists of annotations from a subset of dev and test videos from DiDeMo, Oops, and UVO. The annotated videos were selected based on a few criteria: (i) public availability as of February 22, 2023; (ii) lack of inappropriate content; (iii) annotation quality insurance; (iv) preprocessing criteria (*e.g.*, by removing videos whose first action last less than 1.5s).

**Q8 What data does each instance consist of?** *"Raw" data (e.g., unprocessed text or images) or features? In either case, please provide a description.*

We provide raw annotations and corresponding video filenames (text). In addition, we also release the features used to compute our set of automatic metrics for the ground-truth videos.

**Q9 Is there a label or target associated with each instance?** *If so, please provide a description.*

The goal of the dataset is not to classify any given instance. However, we enrich the annotation of each action to easily analyze failure modes by collecting 35 labels across 6 categories (camera movements, foreground entities, foreground actions, background actions, foreground interactions, foreground transitions). We provide the full list of labels in the main body of the paper.

**Q10 Is any information missing from individual instances?** *If so, please provide a description, explaining why this information is missing (e.g., because it was unavailable). This does not include intentionally removed information, but might include, e.g., redacted text.*

No.

**Q11 Are relationships between individual instances made explicit (e.g., users' movie ratings, social network links)?** *If so, please describe how these relationships are made explicit.*

No.

Q12 **Are there recommended data splits (e.g., training, development/validation, testing)?** *If so, please provide a description of these splits, explaining the rationale behind them.*

Yes. We collect annotations for the existing dev and test splits. We thus recommend using the original training/dev/test splits to avoid any leakage.

Q13 **Are there any errors, sources of noise, or redundancies in the dataset?** *If so, please provide a description.*

Some videos have multiple stories, which correspond to different instances in our datasets. For data collected in VidLN [20], these correspond to descriptions centered around different actors.

Q14 **Is the dataset self-contained, or does it link to or otherwise rely on external resources (e.g., websites, tweets, other datasets)?** *If it links to or relies on external resources, a) are there guarantees that they will exist, and remain constant, over time; b) are there official archival versions of the complete dataset (i.e., including the external resources as they existed at the time the dataset was created); c) are there any restrictions (e.g., licenses, fees) associated with any of the external resources that might apply to a future user? Please provide descriptions of all external resources and any restrictions associated with them, as well as links or other access points, as appropriate.*

Our benchmark and the datasets we collected rely on existing video datasets (DiDeMo, Oops, and UVO). We do not provide archival versions of the complete datasets, but the corresponding video resources are publicly available for download from their official websites.

Q15 **Does the dataset contain data that might be considered confidential (e.g., data that is protected by legal privilege or by doctor–patient confidentiality, data that includes the content of individuals' non-public communications)?** *If so, please provide a description.*
No.

Q16 **Does the dataset contain data that, if viewed directly, might be offensive, insulting, threatening, or might otherwise cause anxiety?** *If so, please describe why.*

Our collected annotations have been verified by humans not to contain inappropriate content. Moreover, our annotators flagged videos that contained sensitive data, which were then all discarded. While we did make an attempt to remove inappropriate content, we cannot exclude that a small number of inappropriate samples might have gone unnoticed.

Q17 **Does the dataset relate to people?** *If not, you may skip the remaining questions in this section.*

Several of our descriptions and corresponding videos are about people. All of the datasets have been verified for sensitive content, and several instances do not include people.

Q18 **Does the dataset identify any subpopulations (e.g., by age, gender)?**

We do not explicitly collect annotations for any subpopulation. However, it may still be possible to deduce this information from the videos and/or the written descriptions.

Q19 **Is it possible to identify individuals (i.e., one or more natural persons), either directly or indirectly (i.e., in combination with other data) from the dataset?** *If so, please describe how.*

Yes, it may be possible to identify people from the videos corresponding to our annotations.

Q20 **Does the dataset contain data that might be considered sensitive in any way (e.g., data that reveals racial or ethnic origins, sexual orientations, religious beliefs, political opinions or union memberships, or locations; financial or health data; biometric or genetic data; forms of government identification, such as social security numbers; criminal history)?** *If so, please provide a description.*

Yes, our data might be considered sensitive. For instance, the associated videos reveal racial or ethnic origins of people shown in them. However, we note that we removed any videos that were found inappropriate by our annotators.

Q21 **Any other comments?**

We call for responsible usage of our datasets for research purposes *only* given the potential of text-guided video generation technologies to affect users.

**Collection Process**

Q22 **How was the data associated with each instance acquired?** *Was the data directly observable (e.g., raw text, movie ratings), reported by subjects (e.g., survey responses), or indirectly*

*inferred/derived from other data (e.g., part-of-speech tags, model-based guesses for age or language)? If data was reported by subjects or indirectly inferred/derived from other data, was the data validated/verified? If so, please describe how.*

We collected human annotations from existing, publicly available video datasets. During the collection campaign, our annotators directly looked at the raw videos. A random sample of the annotations were verified by other humans to ensure high-quality standards.

**Q23  What mechanisms or procedures were used to collect the data (e.g., hardware apparatus or sensor, manual human curation, software program, software API)?** *How were these mechanisms or procedures validated?*

We collected human annotations through web user interfaces that we developed. They were validated by manual inspection by us and managers from the company we hired to collect human annotations.

**Q24  If the dataset is a sample from a larger set, what was the sampling strategy (e.g., deterministic, probabilistic with specific sampling probabilities)?**

We annotated all the videos from the evaluation sets of existing datasets that were still available online at the time of our data collection. We also discarded any videos that were found inappropriate.

**Q25  Who was involved in the data collection process (e.g., students, crowdworkers, contractors) and how were they compensated (e.g., how much were crowdworkers paid)?**

We hired a third-party company to collect human annotations from contractors, who received their standard contracted wage, which is above the living wage in their country of employment. The first and last author were also closely involved during the data collection to ensure that the instructions were clear and resolve any doubts raised by the crowdworkers.

**Q26  Over what timeframe was the data collected? Does this timeframe match the creation timeframe of the data associated with the instances (e.g., recent crawl of old news articles)?** *If not, please describe the timeframe in which the data associated with the instances was created.*

Our video annotations were collected between December 2022 and March 2023, but the corresponding videos were previously collected by other authors.

**Q27  Were any ethical review processes conducted (e.g., by an institutional review board)?** *If so, please provide a description of these review processes, including the outcomes, as well as a link or other access point to any supporting documentation.*

No institutional review board conducted any ethical review process since we do not modify the original videos, and the datasets providing the videos are publicly available and have previously been published in peer-reviewed journals and conferences.

**Q28  Does the dataset relate to people?** *If not, you may skip the remaining questions in this section.*

Yes, people may appear in our annotations as well as in the corresponding videos.

**Q29  Did you collect the data from the individuals in question directly, or obtain it via third parties or other sources (e.g., websites)?**

We collected annotations from crowdworkers, not from the individuals shown in the original videos.

**Q30  Were the individuals in question notified about the data collection?** *If so, please describe (or show with screenshots or other information) how notice was provided, and provide a link or other access point to, or otherwise reproduce, the exact language of the notification itself.*

Individuals were not notified about our data collection, which involved describing their actions in publicly released videos.

**Q31  Did the individuals in question consent to the collection and use of their data?** *If so, please describe (or show with screenshots or other information) how consent was requested and provided, and provide a link or other access point to, or otherwise reproduce, the exact language to which the individuals consented.*

We collect annotations from existing, publicly available video datasets. We do not, however, annotate videos that were no longer available online at the time our annotation campaign was conducted, to adhere with the users' intent to remove their content online.

**Q32  If consent was obtained, were the consenting individuals provided with a mechanism to revoke their consent in the future or for certain uses?** *If so, please provide a description, as well as a link or other access point to the mechanism (if appropriate).*

Users can check whether any of their videos is used in our datasets from the corresponding URLs. If users wish to remove their videos after finding them sensitive, they can contact the hosting party

and request to delete the content from the underlying website. Users can also contact us to request removal of the instances in our datasets corresponding to their videos.

**Q33 Has an analysis of the potential impact of the dataset and its use on data subjects (e.g., a data protection impact analysis) been conducted?** *If so, please provide a description of this analysis, including the outcomes, as well as a link or other access point to any supporting documentation.*

The goal of our datasets is to encourage research towards generative models that can assist and boost artists in generating novel content. However, the resulting technologies could be used to create misinformation online, such as through deepfakes. Yet, we believe that our datasets are the first of their kind to study the problem of generating videos from captions that vary over time. Hence, considering both limitations and opportunities offered by our data, we authorize the dataset for purely academic endeavors.

**Q34 Any other comments?**
No.

**Preprocessing, Cleaning, and/or Labeling**

**Q35 Was any preprocessing/cleaning/labeling of the data done (e.g., discretization or bucketing, tokenization, part-of-speech tagging, SIFT feature extraction, removal of instances, processing of missing values)?** *If so, please provide a description. If not, you may skip the remainder of the questions in this section.*

Yes, we ask human annotators to select which of 34 labels are related to any video segment and captions. We remove any instances (i) whose first action lasts less than 1.5s, or (ii) have a timestamp gap longer than 0.5s between any two consecutive actions.

**Q36 Was the "raw" data saved in addition to the preprocessed/cleaned/labeled data (e.g., to support unanticipated future uses)?** *If so, please provide a link or other access point to the "raw" data.*

No, we do not save the raw data due to data retention policies in our organization.

**Q37 Is the software used to preprocess/clean/label the instances available?** *If so, please provide a link or other access point.*

Yes, we release our preprocessing scripts on GitHub.

**Q38 Any other comments?**
No.

**Uses**

**Q39 Has the dataset been used for any tasks already?** *If so, please provide a description.*

Our datasets have not been used for other tasks yet. However, the underlying videos have been used for their original tasks, such as temporal localization with DiDeMo, studying unintentional human action with Oops, and dense, open-world segmentation with UVO. Moreover, VidLN annotations have been used for the tasks of video narrative grounding and video question answering.

**Q40 Is there a repository that links to any or all papers or systems that use the dataset?** *If so, please provide a link or other access point.*

We encourage the community to measure progress in our benchmark and datasets at the URL `https://paperswithcode.com/dataset/storybench`.

**Q41 What (other) tasks could the dataset be used for?**

Our data can be used for the dual task of describing videos over time. In addition, our data could be used to develop automatic evaluation metrics that better align with human preferences. We also believe that the richness of our data will encourage future work to create new, exciting tasks.

**Q42 Is there anything about the composition of the dataset or the way it was collected and preprocessed/cleaned/labeled that might impact future uses?** *For example, is there anything that a future user might need to know to avoid uses that could result in unfair treatment of individuals or groups (e.g., stereotyping, quality of service issues) or other undesirable harms (e.g., financial harms, legal risks) If so, please provide a description. Is there anything a future user could do to mitigate these undesirable harms?*

Our annotations describe existing video datasets that might not contain a fair distribution of individuals or groups. For our task, this means that models might be able to generate videos that are biased

towards the populations represented in the training data. We encourage future work to extend our efforts towards creating training and evaluation datasets that specifically aim to increase fairness and reduce biases (*e.g.*, correlation between gender, race and jobs) of generative text-to-video models.

**Q43  Are there tasks for which the dataset should not be used?** *If so, please provide a description.*

Under no circumstances should any models developed for our benchmark be used to create deepfakes or any other form of disinformation or harm, including military and surveillance tasks. As it stands, our datasets should solely be used for research purposes.

**Q44  Any other comments?**

No.

**Distribution**

**Q45  Will the dataset be distributed to third parties outside of the entity (e.g., company, institution, organization) on behalf of which the dataset was created?** *If so, please provide a description.*

Yes, the data will be publicly released.

**Q46  How will the dataset be distributed (e.g., tarball on website, API, GitHub)?** *Does the dataset have a digital object identifier (DOI)?*

The data will be available on GitHub.

**Q47  When will the dataset be distributed?**

From September 2023 and onward.

**Q48  Will the dataset be distributed under a copyright or other intellectual property (IP) license, and/or under applicable terms of use (ToU)?** *If so, please describe this license and/or ToU, and provide a link or other access point to, or otherwise reproduce, any relevant licensing terms or ToU, as well as any fees associated with these restrictions.*

CC-BY-4.0

**Q49  Have any third parties imposed IP-based or other restrictions on the data associated with the instances?** *If so, please describe these restrictions, and provide a link or other access point to, or otherwise reproduce, any relevant licensing terms, as well as any fees associated with these restrictions.*

No, our collected annotations are released under a CC-BY-4.0 license. Third-party data are also released publicly.

**Q50  Do any export controls or other regulatory restrictions apply to the dataset or to individual instances?** *If so, please describe these restrictions, and provide a link or other access point to, or otherwise reproduce, any supporting documentation.*

No.

**Q51  Any other comments?**

No.

**Maintenance**

**Q52  Who will be supporting/hosting/maintaining the dataset?**

Google Research will support and maintain the STORYBENCH annotations on GitHub. The original videos are supported by the corresponding dataset creators or services.

**Q53  How can the owner/curator/manager of the dataset be contacted (e.g., email address)?**

We can be contacted either via email or through 'pull requests' on the STORYBENCH GitHub page.

**Q54  Is there an erratum?** *If so, please provide a link or other access point.*

There is no erratum for our first release. Errata will be documented as future releases on GitHub.

**Q55  Will the dataset be updated (e.g., to correct labeling errors, add new instances, delete instances)?** *If so, please describe how often, by whom, and how updates will be communicated to users (e.g., mailing list, GitHub)?*

No, we do not plan on updating the data. However, we will update the data should there be any errors or requests for deleting specific instances. The updated data will be shared as a 'release' on GitHub.

**Q56** **If the dataset relates to people, are there applicable limits on the retention of the data associated with the instances (e.g., were individuals in question told that their data would be retained for a fixed period of time and then deleted)?** *If so, please describe these limits and explain how they will be enforced.*

We do not collect any metadata related to people in creating STORYBENCH. However, we note that our datasets consist of text annotations of existing video datasets. Should people request for their videos to be deleted from the original datasets, we invite them and users to contact us to ensure that the corresponding annotations are removed from STORYBENCH.

**Q57** **Will older versions of the dataset continue to be supported/hosted/maintained?** *If so, please describe how. If not, please describe how its obsolescence will be communicated to users.*

Yes, we will distribute all versions of STORYBENCH as 'releases' on GitHub.

**Q58** **If others want to extend/augment/build on/contribute to the dataset, is there a mechanism for them to do so?** *If so, please provide a description. Will these contributions be validated/verified? If so, please describe how. If not, why not? Is there a process for communicating/distributing these contributions to other users? If so, please provide a description.*

There is no plan to support and verify third-party contributions that aim at extending the datasets in STORYBENCH as our annotations correspond to standard evaluation splits of existing video datasets. However, we will update the data should there be any errors or requests for deleting specific instances. Dataset versions will be maintained as GitHub releases.

**Q59** **Any other comments?**
No.

# D  Data Preparation Details

In this section, we provide further details on the preparation pipeline used for the evaluation (dev and test) data in STORYBENCH. It consists of the following steps: collection, preprocessing, and rating.

## D.1  Data Collection

First, we use an online interface (see Figure 5) to collect stories for each video. Stories consist of multiple sentences, each describing an action, and the corresponding timestamps in the video.

**Oops-CSV and UVO-CSV.** For VidLN data (UVO and Oops), we provide the original VidLN caption, as well as reference split captions provided from our automatic pipeline (*c.f.* Section 4). In this stage, annotators were instructed to 'split the long sentence into shorter sentences, each describing actions that happen one after the other' and to 'add time stamps for when (the action of) each sentence starts and ends.' Moreover, our annotators are asked to click two checkboxes, whenever applicable: 'Multiple stories,' used to indicate whether the video shown in the user interface actually consists of multiple shorter clips (this is common in Oops, as the data consists of fail video *compilations*); and 'Unimportant actor,' used to indicate whether the original caption describes the events in the video from the perspective of an entity that does not play a salient role in the video (*e.g.*, a person in the background). Finally, we perform a second stage of annotations where we provide annotators with the stories from the first stage, and ask them to 'continue the sentence that describes the actor's action in a natural manner by adding a concise context description of relevant actions of other actors.' This second round of annotations was required as VidLN describes the actions of a given entity (actor) throughout a video, which does not often capture the dynamics of the corresponding video segments. Our annotators narrate 2,446 and 2,779 videos from the dev sets of Oops and UVO, respectively.

**DiDeMo-CSV.** For DiDeMo, we do not possess any descriptions of the video. Instead, the dataset provides detailed text queries (*e.g.*, containing camera movement, temporal transition indicators, and activities) that are used to localize events in the video. We provide those queries as a reference to our annotators, and ask them to 'add a description of the background,' 'specify the number of important actors in the video,' 'refine the sentences to create a coherent story,' and 'add timestamps for when (the action of) each sentence starts and ends.' Following the original DiDeMo protocol, each video is trimmed to a maximum of 30 seconds. While the original dev and test sets contained 1,065 and 1,004 videos, respectively, only 843 and 797 videos were still publicly available as of February 22, 2023.

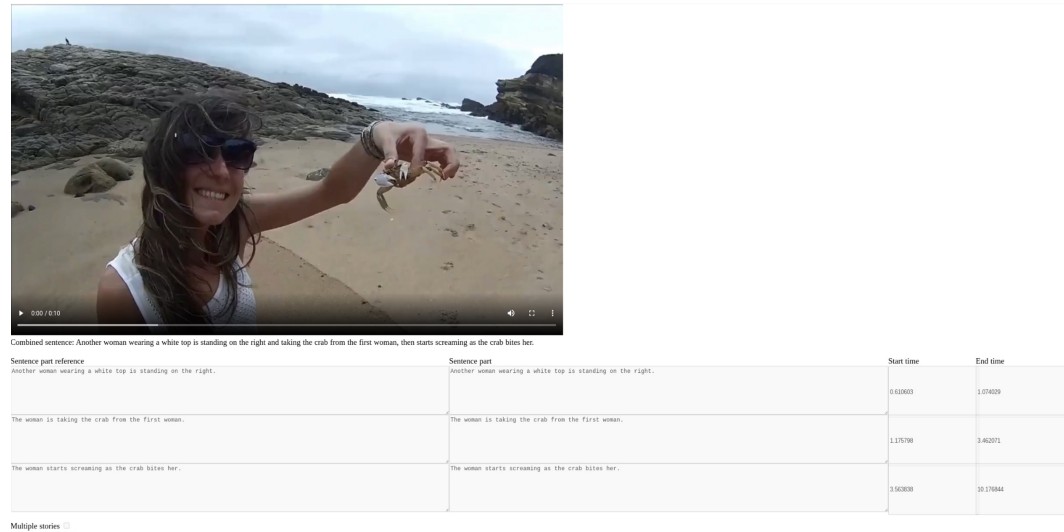

Figure 5: Example of our video annotation interface.

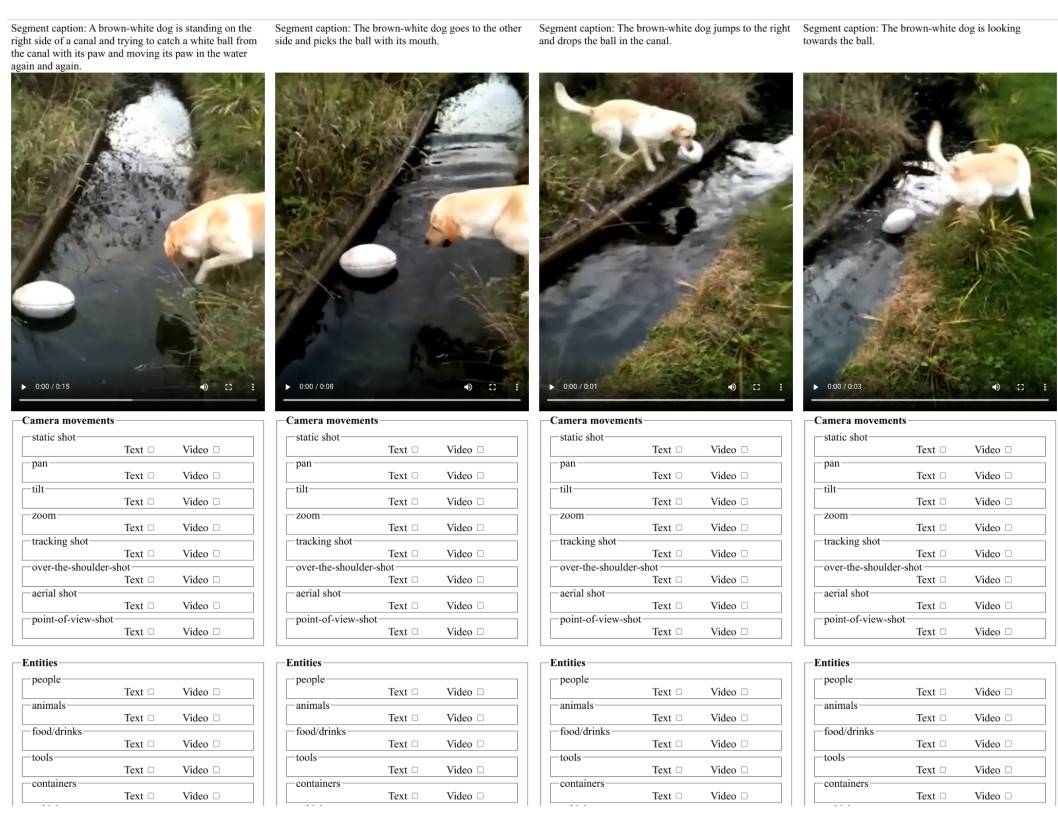

Figure 6: Example of our diagnostic labels collection interface.

In both cases, we provide additional details for both of these tasks to the annotators, as well as examples and corner cases to clearly communicate the desired annotations (available on GitHub). During this collection process, any video flagged by the annotators to contain inappropriate content was removed. Finally, a random sample of our annotations were verified by expert annotators identified by the third party company responsible for human annotations in this project.

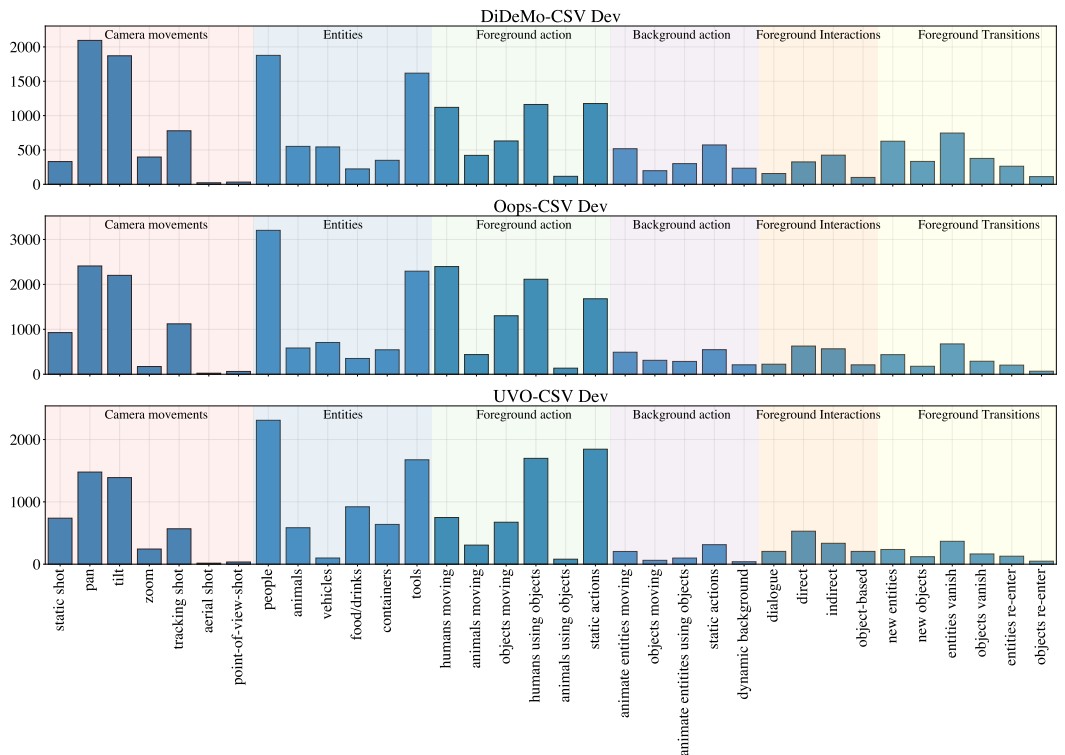

Figure 7: Distribution of collected labels per category in our dev samples.

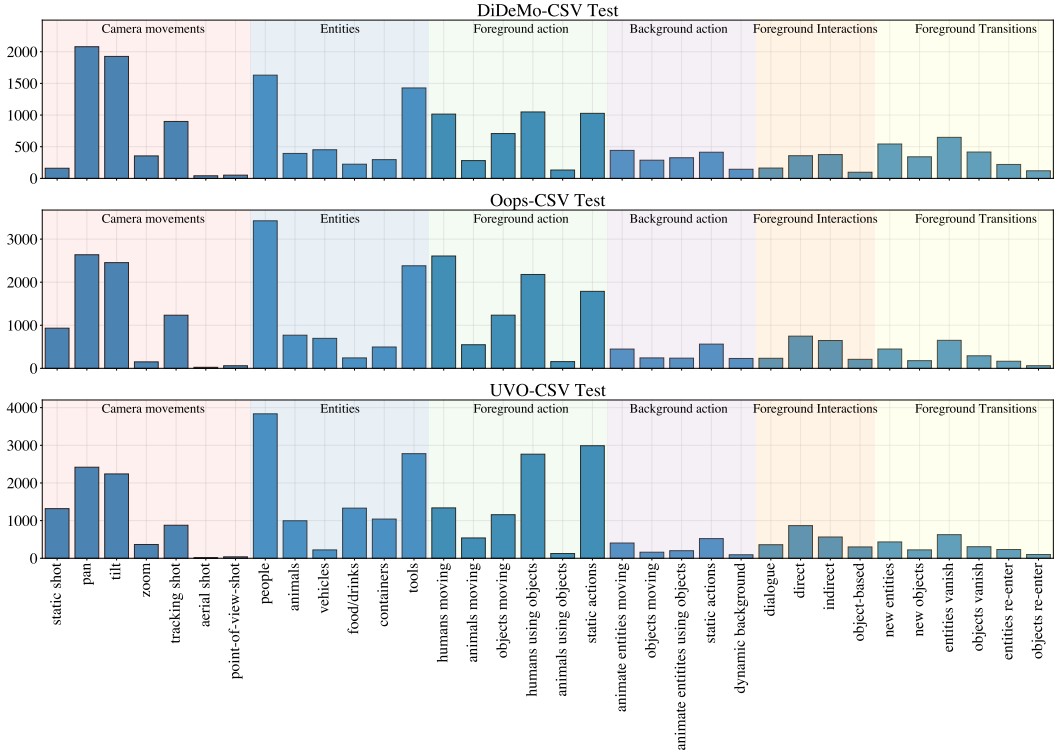

Figure 8: Distribution of collected labels per category in our test samples.

**Diagnostic labels.**   After collecting story annotations for our videos, we enrich them with labels to help analyze the performance of forthcoming text-to-video models along different axes. With the help of artists that have been using generative AI technologies, we define 34 labels across six categories (*c.f*. Section 3). For each video segment, we then ask our annotators to tick two checkboxes per label: 'Text' if the label is mentioned in the segment caption; and 'Video' if the label is shown in the video. Figure 6 shows an example of the UI used in this process, and we release our full set of instructions online. Figures 7 and 8 show the distribution of labels in our preprocessed (*i.e.*, final) data.

**Human annotation framework.**   We assess the high quality of our annotations as follows. First, annotators were only moved to the final data annotation process after having successfully completed a training stage. Second, we asked annotators' managers to verify the quality (*i.e.*, descriptions and timestamps match the content of the videos) of the final data by manually checking 25% of the data samples. Here, they found 97% of the samples to accurately reflect the narratives of the videos.

## D.2   Data Preprocessing

Given the above collected annotations, we perform the following two preprocessing steps. First, we only keep stories whose first action is at least 1.5s long; as we use a video of 0.5s to condition text-to-video generation for the task of *story continuation*. Second, we remove any story in which two subsequent actions have more than 0.5s gap.

Figures 9 and 10 shows our final data distributions. For DiDeMo-CSV, the dev split has 744/744 videos/stories, while the test split has 655/655 videos/stories. For Oops-CSV, the dev split has 979/1578 videos/stories, while the test split has 979/1578 videos/stories. For UVO-CSV, the dev split has 1019/1665 videos/stories, while the test split has 1565/2613 videos/stories.

The preprocessed data is then adapted for each of our evaluation tasks detailed in Section 3: *action execution*, *story continuation*, and *story generation*. Finally, to evaluate our baselines, the original videos are downsampled to 8 frames per second (fps) using the 'FFmpeg' open-source software.

Figures 12 to 14 show examples of the resulting data.

## D.3   Human Evaluation

Human evaluation is the preferred way to assess the capabilities of generative models. We perform side-by-side comparisons between two models, and ask human raters to choose the one (if any) that performs better according to the five criteria that we defined in Section 3. Figure 15 shows an example of the user interface developed for human evaluation.

## D.4   Automatic Evaluation

Section 3.4 introduces our automatic evaluation metrics. Here, we provide our intuition of how we expect them to relate to our human evaluation metrics. **FID** would measure "visual quality" since it compares the distribution of ground-truth frames with that of generated frames. **FVD** would measure "entity consistency" and "action realism" since it compares the distribution of ground-truth videos with that of generated videos. **SIM** would measure "visual quality", "entity consistency", "background consistency", and "text adherence" as it compares ground-truth and generated frames one-to-one. **VTM** would measure "text adherence" as it compares generated videos to their prompts. **PQA** would measure "visual quality" and "action realism" as it was trained to predict the average human subjective perception of a video.

## D.5   Robustness of Automatic Pipeline

In Section 4, we define an automatic pipeline to transform the original VidLN captions for Oops and UVO into multiple sentences, each approximately describing a single action, and to estimate their corresponding timestamps. Here, we compute some statistics to assess the quality of the stories generated automatically through our algorithmic pipeline by comparing them against human references for the Oops Dev set (1,578 stories).

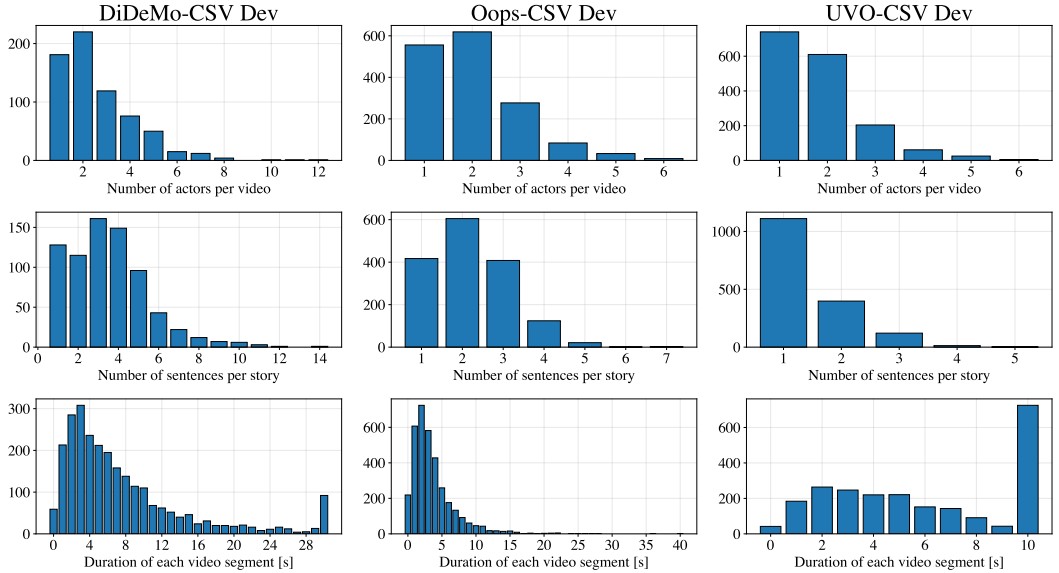

Figure 9: Statistics of our final dev sets.

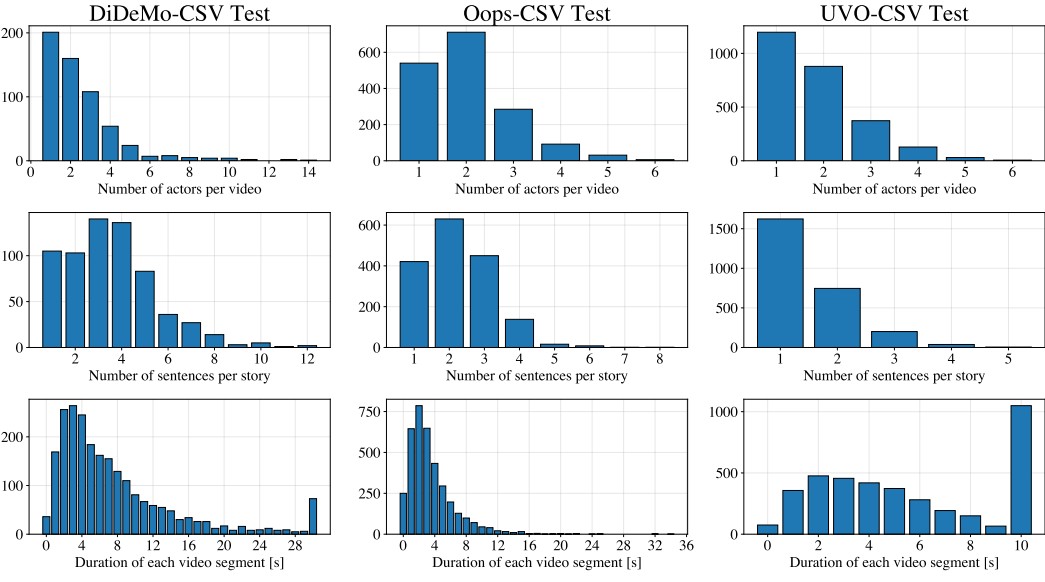

Figure 10: Statistics of our final test sets.

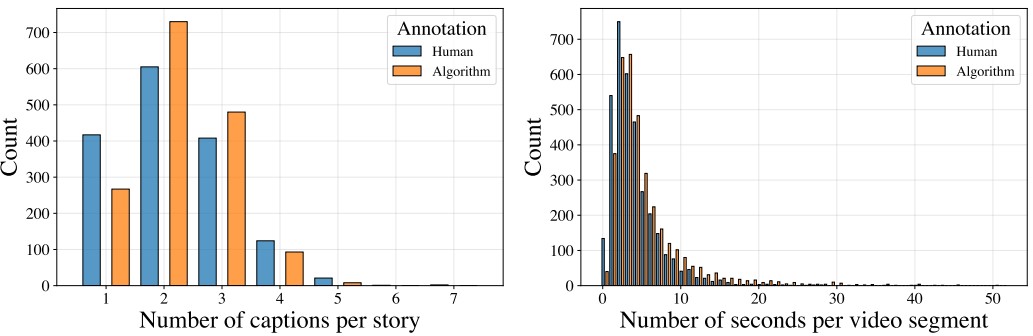

Figure 11: Robustness statistics of automated pipeline for story-like data transformation. Left: Number of captions per story. Right: Duration of video segments in seconds.

PROMPT: A man wearing white shorts is jumping on a trampoline.

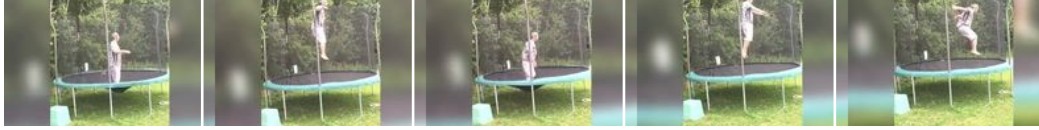

PROMPT: The man performing a flip.

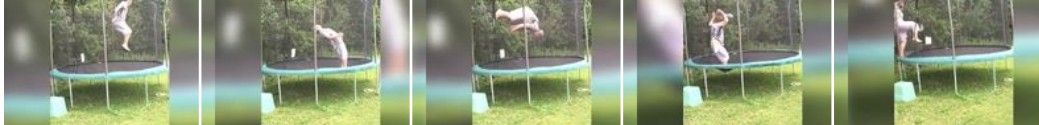

PROMPT: The man falls when the trampoline falls on the ground.

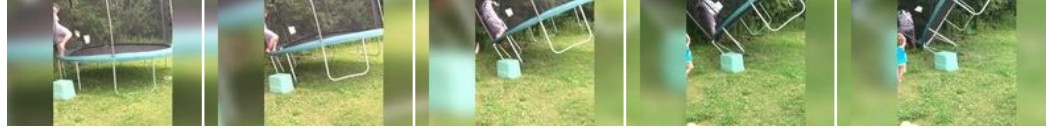

Figure 12: Example story (subsampled frames) from Oops-CSV.

PROMPT: A baby wearing blue clothes first touches the girl's ice cream while the first girl is eating her ice cream.

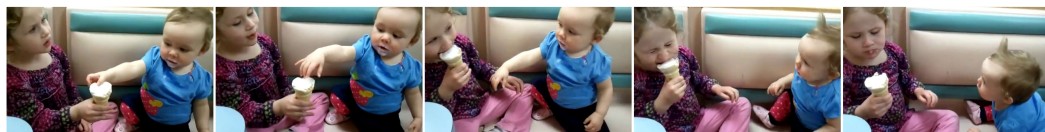

PROMPT: The baby turns back.

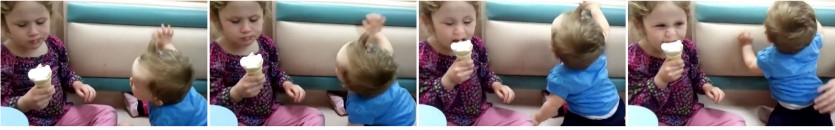

PROMPT: The baby starts climbing on the back side of the seat.

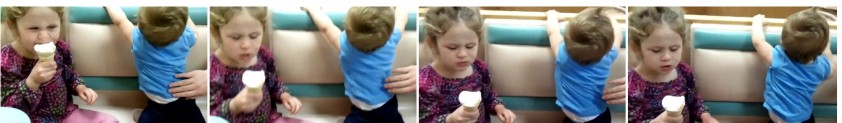

Figure 13: Example story (subsampled frames) from UVO-CSV.

As shown in Figure 11 (left), the distributions of the number of sentences per story of the two approaches are very similar. In particular, we notice that our method tends to split captions into two or three segments more often than the human annotators, who, more often, prefer not to split them.

The words corresponding to each video segment are very similar between human and automatic stories. To assess this, we consider the subset of captions that have been split into the same number of sentences, so we can compute one-to-one mappings between the human and the algorithmic captions. Here, we observe a BLEU$_4$ score of 63.6% (BLEU$_4$ measures the overlap of 4-grams in the two captions), indicating a relatively high similarity of generated sentences to human references. We also note that humans were asked to enrich the original Oops and UVO captions with context information (*e.g.*, what other relevant actors are doing while a specific actor is being narrated), which our algorithmic pipeline does not explicitly tackle, leaving room for improvement in future work.

Finally, Figure 11 (right) shows that the resulting duration of the algorithmically generated video segments are slightly longer than human-annotated timestamps.

PROMPT: A white-brown dog is sitting and starts moving towards the person.

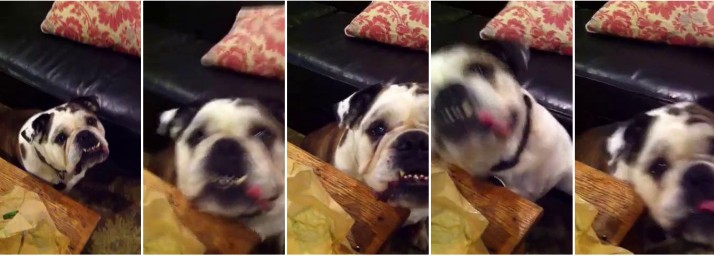

PROMPT: A person whose only hand and leg is visible is holding some food in his hand.

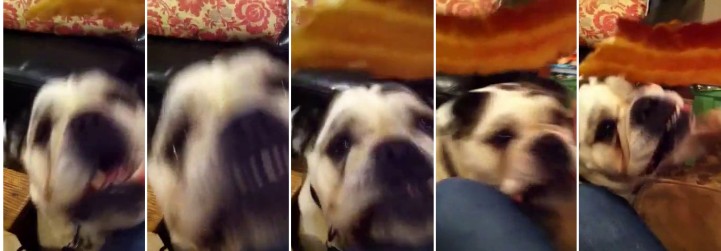

PROMPT: The white-brown dog is take food from the person hand and eats, while the camera focus on the dog face.

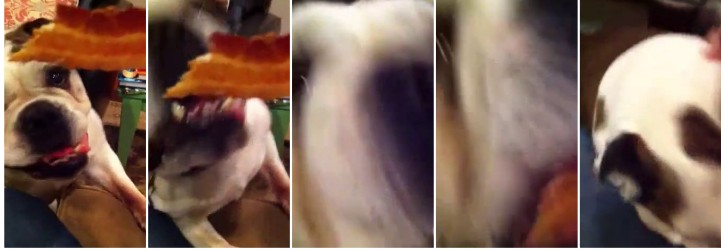

PROMPT: The person starts rubbing the dog head with his hand.

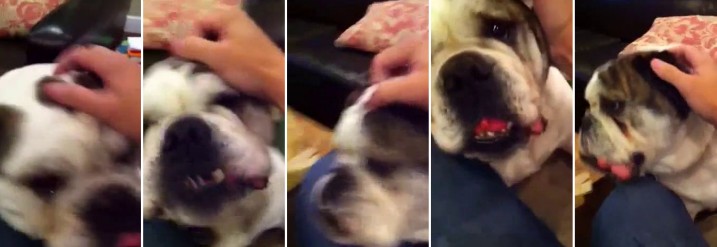

Figure 14: Example story (subsampled frames) from DiDeMo-CSV.

# E    Additional Results

In this section, we report our full set of results from our baselines on STORYBENCH, in terms of both human evaluations and through automatic metrics. Recall that we append -ZS for results obtained in the *zero-shot* setting, -ST for *single-task* fine-tuning, and -MT for *multi-task* fine-tuning. Each model was fine-tuned for 500K steps in less than a day on 4x4x4 TPUv4 chips. For every story, each model generates 4 output videos at 8fps using a 160×96 pixel resolution. We randomly sample one of them for human evaluation (*e.g.*, Figure 16), but report mean and standard deviation for automatic metrics.

## E.1    Human Evaluation

Figure 17 shows the results of human evaluation, where each bar displays the number of wins of two given models evaluated side-by-side, as well as the number of ties (in white). For each story, we ask three human raters to compare two models and report the majority vote in Figure 17.

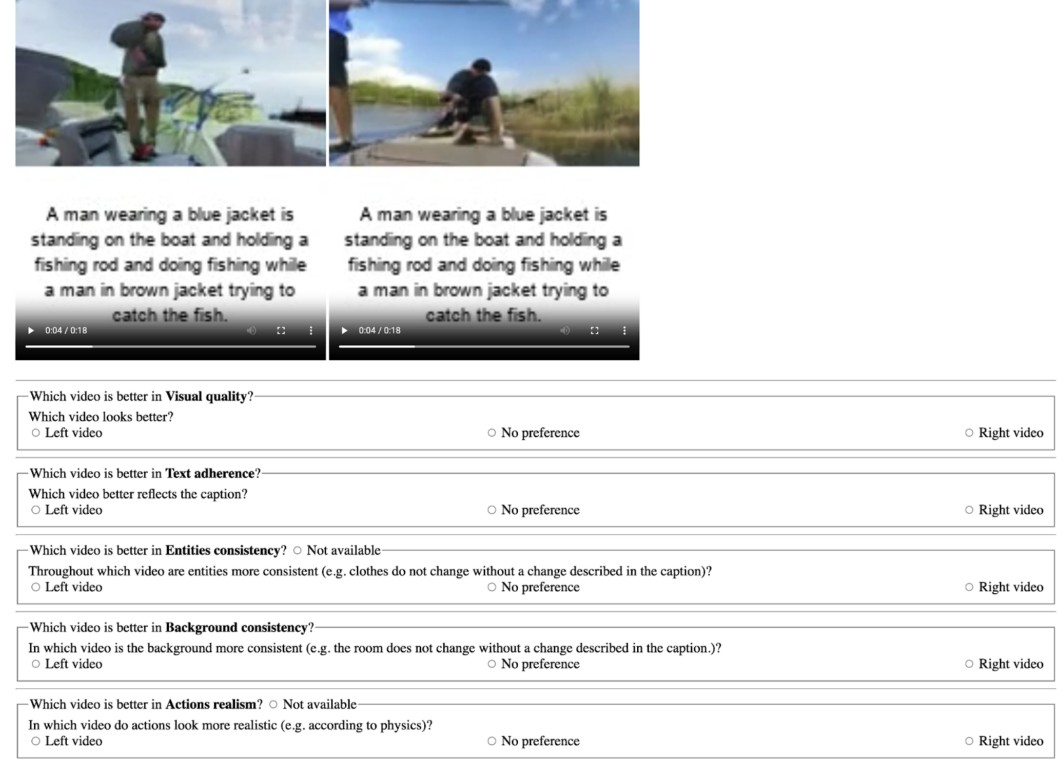

Figure 15: Example of our human rating interface.

PROMPT: The swimmers dive into the water and starts swimming from one end to the another.

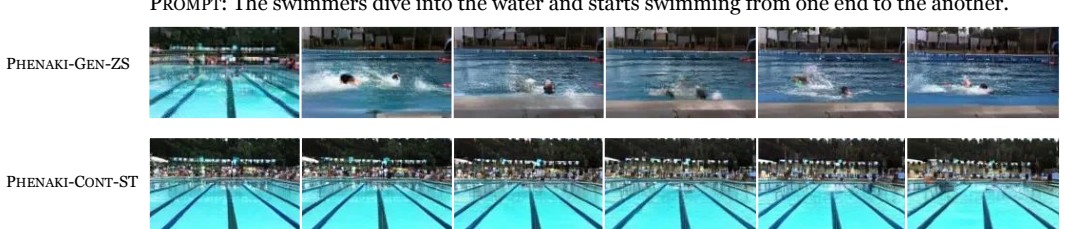

Figure 16: Example of generated actions by PHENAKI-GEN-ST and PHENAKI-CONT-ST on DiDeMo-CSV. PHENAKI-GEN-ST quickly changes the background, while PHENAKI-CONT-ST correctly synthesizes a person swimming left-to-right without distorting the background. Video subsampled by a factor 4 to be shown here.

Looking at task of *action execution* on Oops-CSV, we see that our PHENAKI-CONT-ST achieves competitive performance with our PHENAKI-GEN-ZS baseline, with better text adherence, background consistency and action realism. This result is not surprising as most of the actions in Oops are short (less than 5s). It is interesting, however, to see that our annotators find PHENAKI-CONT-ST largely better than PHENAKI-GEN-ST across all criteria. On the other hand, none of these models clearly outperforms others for the most challenging task of *story generation*.

For the task of *story continuation*, PHENAKI-CONT-ST typically outperforms both PHENAKI-GEN-ZS and PHENAKI-GEN-ST, especially on Oops-CSV. On UVO-CSV and DiDeMo-CSV, PHENAKI-CONT-ST consistently outperforms PHENAKI-GEN-ST except for entity and background consistency, where human raters often have no preference between the two. Comparing multi-task models, we find that PHENAKI-CONT-MT is always preferred to PHENAKI-GEN-MT; yet PHENAKI-GEN-ZS is a strong baseline, achieving better visual quality than the fine-tuned models.

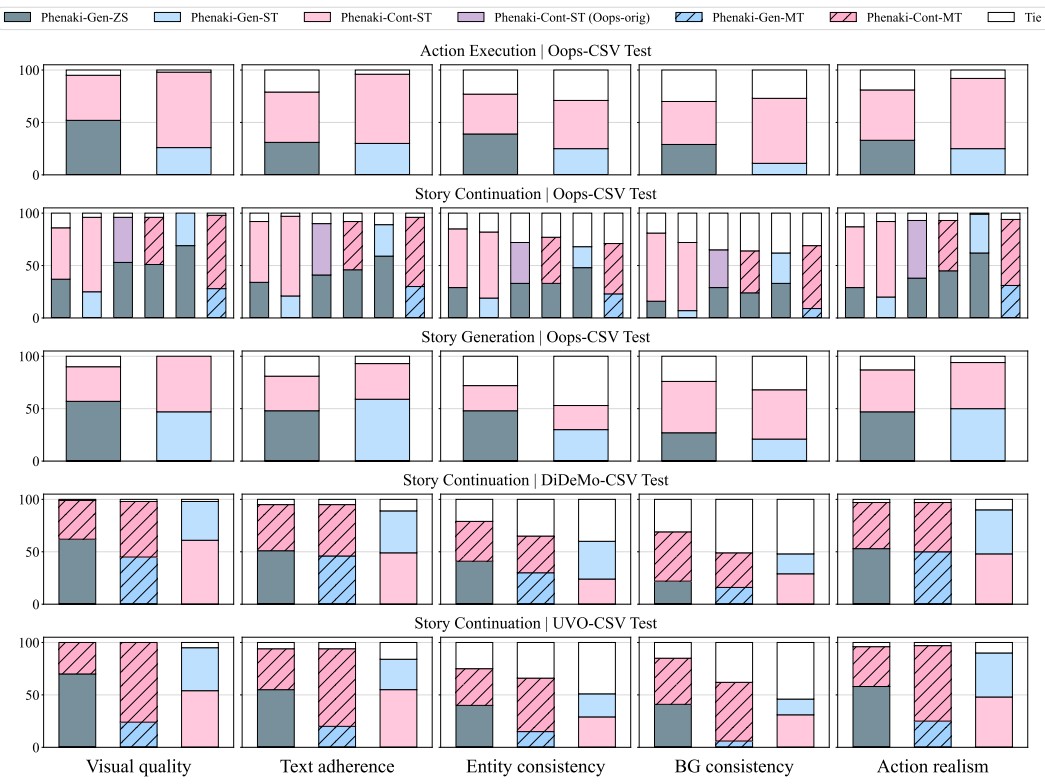

Figure 17: Results from human evaluation across datasets and tasks.

## E.2 Automatic Evaluation

For completeness, Tables 8 to 10 report the performance of our baselines on all tasks and datasets when instead using CLIP to compute FID and SIM, and InternVideo to compute FVD and VTM. We find similar patterns as with other metrics (*c.f.* Section 6), but also notice that InternVideo (used for FVD and VTM) favors the videos generated by the zero-shot PHENAKI-GEN model.

**Table 8 — Action Execution**

| Action Execution
Model (@8 fps) | Oops-CSV | | | | | UVO-CSV | | | | | DiDeMo-CSV | | | | |
|---|---|---|---|---|---|---|---|---|---|---|---|---|---|---|---|
| | $FID_C\downarrow$ | $FVD_{IV}\downarrow$ | $SIM_C\uparrow$ | $PQA\uparrow$ | $VTM_{IV}\uparrow$ | $FID_C\downarrow$ | $FVD_{IV}\downarrow$ | $SIM_C\uparrow$ | $PQA\uparrow$ | $VTM_{IV}\uparrow$ | $FID_C\downarrow$ | $FVD_{IV}\downarrow$ | $SIM_C\uparrow$ | $PQA\uparrow$ | $VTM_{IV}\uparrow$ |
| *Zero-shot* | | | | | | | | | | | | | | | |
| PHENAKI-GEN-ZS | $94.7_{\pm0.52}$ | $\mathbf{126.7}_{\pm0.46}$ | $64.9_{\pm0.08}$ | $\mathbf{5.8}_{\pm0.03}$ | $\mathbf{22.6}_{\pm0.07}$ | $79.2_{\pm0.38}$ | $\mathbf{85.3}_{\pm0.41}$ | $66.7_{\pm0.03}$ | $\mathbf{8.5}_{\pm0.10}$ | $\mathbf{23.0}_{\pm0.03}$ | $97.2_{\pm0.34}$ | $\mathbf{78.0}_{\pm0.25}$ | $64.3_{\pm0.08}$ | $\mathbf{6.7}_{\pm0.02}$ | $\mathbf{22.9}_{\pm0.05}$ |
| *Single-Task* | | | | | | | | | | | | | | | |
| PHENAKI-GEN-ST | $97.1_{\pm0.20}$ | $179.4_{\pm0.28}$ | $64.8_{\pm0.04}$ | $4.0_{\pm0.02}$ | $20.0_{\pm0.02}$ | $97.3_{\pm0.18}$ | $147.6_{\pm0.20}$ | $62.5_{\pm0.04}$ | $4.8_{\pm0.05}$ | $18.4_{\pm0.02}$ | $89.3_{\pm0.22}$ | $120.2_{\pm0.59}$ | $64.9_{\pm0.04}$ | $4.5_{\pm0.02}$ | $20.4_{\pm0.03}$ |
| PHENAKI-CONT-ST | $\mathbf{84.5}_{\pm0.02}$ | $171.6_{\pm0.58}$ | $\mathbf{67.9}_{\pm0.04}$ | $4.8_{\pm0.02}$ | $19.9_{\pm0.01}$ | $92.4_{\pm0.27}$ | $143.2_{\pm0.31}$ | $64.0_{\pm0.09}$ | $5.6_{\pm0.02}$ | $18.7_{\pm0.03}$ | $\mathbf{82.5}_{\pm0.22}$ | $107.3_{\pm0.46}$ | $\mathbf{66.8}_{\pm0.01}$ | $5.6_{\pm0.01}$ | $20.1_{\pm0.01}$ |
| *Multi-Task* | | | | | | | | | | | | | | | |
| PHENAKI-GEN-MT | $102.8_{\pm0.64}$ | $179.3_{\pm0.63}$ | $63.7_{\pm0.04}$ | $3.8_{\pm0.04}$ | $20.1_{\pm0.04}$ | $92.1_{\pm0.68}$ | $138.9_{\pm0.42}$ | $63.3_{\pm0.03}$ | $5.1_{\pm0.05}$ | $19.2_{\pm0.02}$ | $88.2_{\pm0.44}$ | $119.6_{\pm0.47}$ | $64.3_{\pm0.04}$ | $4.7_{\pm0.06}$ | $20.1_{\pm0.02}$ |
| PHENAKI-CONT-MT | $86.0_{\pm0.52}$ | $171.3_{\pm0.56}$ | $67.4_{\pm0.11}$ | $4.7_{\pm0.01}$ | $20.1_{\pm0.03}$ | $\mathbf{77.9}_{\pm0.08}$ | $126.8_{\pm0.24}$ | $\mathbf{67.1}_{\pm0.06}$ | $6.8_{\pm0.01}$ | $19.9_{\pm0.02}$ | $85.4_{\pm0.14}$ | $106.5_{\pm0.29}$ | $66.4_{\pm0.07}$ | $5.8_{\pm0.03}$ | $19.9_{\pm0.03}$ |

Table 8: Results from automatic evaluation metrics on *action execution* tasks. Best results are in **bold**. FID and SIM use CLIP, FVD and VTM use InternVideo, and PQA uses DOVER.

**Table 9 — Story Continuation**

| Story Continuation
Model (@8 fps) | Oops-CSV | | | | | UVO-CSV | | | | | DiDeMo-CSV | | | | |
|---|---|---|---|---|---|---|---|---|---|---|---|---|---|---|---|
| | $FID_C\downarrow$ | $FVD_{IV}\downarrow$ | $SIM_C\uparrow$ | $PQA\uparrow$ | $VTM_{IV}\uparrow$ | $FID_C\downarrow$ | $FVD_{IV}\downarrow$ | $SIM_C\uparrow$ | $PQA\uparrow$ | $VTM_{IV}\uparrow$ | $FID_C\downarrow$ | $FVD_{IV}\downarrow$ | $SIM_C\uparrow$ | $PQA\uparrow$ | $VTM_{IV}\uparrow$ |
| *Zero-shot* | | | | | | | | | | | | | | | |
| PHENAKI-GEN-ZS | $103.2_{\pm0.87}$ | $\mathbf{116.6}_{\pm0.54}$ | $63.1_{\pm0.05}$ | $\mathbf{7.2}_{\pm0.06}$ | $\mathbf{22.5}_{\pm0.05}$ | $82.2_{\pm0.51}$ | $\mathbf{83.6}_{\pm0.44}$ | $65.9_{\pm0.04}$ | $\mathbf{9.4}_{\pm0.09}$ | $\mathbf{22.9}_{\pm0.03}$ | $108.2_{\pm0.43}$ | $\mathbf{87.9}_{\pm0.41}$ | $61.7_{\pm0.04}$ | $\mathbf{7.3}_{\pm0.07}$ | $\mathbf{22.5}_{\pm0.10}$ |
| *Single-Task* | | | | | | | | | | | | | | | |
| PHENAKI-GEN-ST | $99.6_{\pm0.07}$ | $181.5_{\pm0.74}$ | $64.0_{\pm0.05}$ | $4.3_{\pm0.02}$ | $19.7_{\pm0.02}$ | $97.8_{\pm0.26}$ | $151.1_{\pm0.21}$ | $62.5_{\pm0.03}$ | $5.0_{\pm0.03}$ | $18.2_{\pm0.01}$ | $90.9_{\pm0.18}$ | $127.5_{\pm0.48}$ | $64.2_{\pm0.03}$ | $4.0_{\pm0.02}$ | $19.9_{\pm0.02}$ |
| PHENAKI-CONT-ST | $\mathbf{89.2}_{\pm0.30}$ | $169.9_{\pm0.67}$ | $\mathbf{66.3}_{\pm0.05}$ | $5.3_{\pm0.04}$ | $19.5_{\pm0.02}$ | $94.1_{\pm0.29}$ | $147.3_{\pm0.75}$ | $63.5_{\pm0.07}$ | $5.7_{\pm0.03}$ | $18.3_{\pm0.02}$ | $\mathbf{89.4}_{\pm0.29}$ | $118.1_{\pm0.60}$ | $\mathbf{64.5}_{\pm0.05}$ | $5.4_{\pm0.03}$ | $19.4_{\pm0.06}$ |
| *Multi-Task* | | | | | | | | | | | | | | | |
| PHENAKI-GEN-MT | $105.2_{\pm0.26}$ | $182.0_{\pm0.17}$ | $63.0_{\pm0.04}$ | $4.2_{\pm0.02}$ | $19.8_{\pm0.02}$ | $92.8_{\pm0.58}$ | $141.9_{\pm0.22}$ | $63.1_{\pm0.03}$ | $5.2_{\pm0.03}$ | $18.9_{\pm0.02}$ | $90.7_{\pm0.24}$ | $126.4_{\pm0.82}$ | $63.4_{\pm0.02}$ | $4.3_{\pm0.04}$ | $19.7_{\pm0.04}$ |
| PHENAKI-CONT-MT | $92.1_{\pm0.44}$ | $171.4_{\pm0.67}$ | $65.7_{\pm0.09}$ | $5.1_{\pm0.02}$ | $19.8_{\pm0.02}$ | $\mathbf{80.4}_{\pm0.17}$ | $129.0_{\pm0.65}$ | $\mathbf{66.3}_{\pm0.09}$ | $7.0_{\pm0.02}$ | $19.6_{\pm0.05}$ | $95.5_{\pm0.33}$ | $120.2_{\pm0.23}$ | $63.4_{\pm0.07}$ | $5.5_{\pm0.09}$ | $19.0_{\pm0.01}$ |

Table 9: Results from automatic evaluation metrics on *story continuation* tasks. Best results are in **bold**. FID and SIM use CLIP, FVD and VTM use InternVideo, and PQA uses DOVER.

**Table 10 — Story Generation**

| Story Generation
Model (@8 fps) | Oops-CSV | | | | | UVO-CSV | | | | | DiDeMo-CSV | | | | |
|---|---|---|---|---|---|---|---|---|---|---|---|---|---|---|---|
| | $FID_C\downarrow$ | $FVD_{IV}\downarrow$ | $SIM_C\uparrow$ | $PQA\uparrow$ | $VTM_{IV}\uparrow$ | $FID_C\downarrow$ | $FVD_{IV}\downarrow$ | $SIM_C\uparrow$ | $PQA\uparrow$ | $VTM_{IV}\uparrow$ | $FID_C\downarrow$ | $FVD_{IV}\downarrow$ | $SIM_C\uparrow$ | $PQA\uparrow$ | $VTM_{IV}\uparrow$ |
| *Zero-shot* | | | | | | | | | | | | | | | |
| PHENAKI-GEN-ZS | $117.2_{\pm0.90}$ | $\mathbf{113.0}_{\pm0.54}$ | N/A | $\mathbf{8.1}_{\pm0.03}$ | $\mathbf{22.9}_{\pm0.11}$ | $97.5_{\pm0.89}$ | $\mathbf{88.2}_{\pm0.68}$ | N/A | $\mathbf{10.0}_{\pm0.06}$ | $\mathbf{22.6}_{\pm0.14}$ | $115.6_{\pm0.38}$ | $\mathbf{91.1}_{\pm0.46}$ | N/A | $\mathbf{7.6}_{\pm0.08}$ | $\mathbf{23.0}_{\pm0.06}$ |
| *Single-Task* | | | | | | | | | | | | | | | |
| PHENAKI-GEN-ST | $\mathbf{103.4}_{\pm0.28}$ | $181.4_{\pm0.19}$ | N/A | $4.2_{\pm0.02}$ | $19.6_{\pm0.03}$ | $99.2_{\pm0.22}$ | $151.7_{\pm0.45}$ | N/A | $4.9_{\pm0.01}$ | $18.0_{\pm0.01}$ | $92.3_{\pm0.07}$ | $128.6_{\pm0.44}$ | N/A | $4.0_{\pm0.01}$ | $20.1_{\pm0.04}$ |
| PHENAKI-CONT-ST | $109.1_{\pm0.89}$ | $167.9_{\pm0.95}$ | N/A | $5.4_{\pm0.03}$ | $17.9_{\pm0.03}$ | $100.6_{\pm0.47}$ | $149.6_{\pm0.24}$ | N/A | $5.4_{\pm0.02}$ | $17.4_{\pm0.02}$ | $96.1_{\pm0.11}$ | $124.6_{\pm0.86}$ | N/A | $5.4_{\pm0.05}$ | $18.9_{\pm0.02}$ |
| *Multi-Task* | | | | | | | | | | | | | | | |
| PHENAKI-GEN-MT | $107.4_{\pm0.71}$ | $180.0_{\pm0.52}$ | N/A | $4.1_{\pm0.02}$ | $19.8_{\pm0.06}$ | $\mathbf{94.8}_{\pm0.25}$ | $143.0_{\pm0.28}$ | N/A | $5.1_{\pm0.03}$ | $18.8_{\pm0.01}$ | $\mathbf{92.0}_{\pm0.20}$ | $127.3_{\pm0.46}$ | N/A | $4.3_{\pm0.06}$ | $19.8_{\pm0.01}$ |
| PHENAKI-CONT-MT | $114.3_{\pm0.12}$ | $171.0_{\pm0.53}$ | N/A | $5.0_{\pm0.11}$ | $18.0_{\pm0.06}$ | $99.8_{\pm0.28}$ | $132.7_{\pm0.52}$ | N/A | $6.3_{\pm0.08}$ | $17.5_{\pm0.05}$ | $105.6_{\pm0.26}$ | $125.9_{\pm1.03}$ | N/A | $5.2_{\pm0.03}$ | $18.3_{\pm0.06}$ |

Table 10: Results from automatic evaluation metrics on *story generation* tasks. Best results are in **bold**. FID uses CLIP, FVD and VTM use InternVideo, and PQA uses DOVER.

