# OpenReview forum: "StoryBench: A Multifaceted Benchmark for Continuous Story Visualization"
_NeurIPS.cc/2023/Track/Datasets_and_Benchmarks — NeurIPS 2023 Datasets and Benchmarks Poster_

### Official Review · Reviewer_i1xQ · 2023-07-06
**This paper presents a benchmark for story generation, consisting of three tasks of increasing difficulty: single-action generation, sequential-action generation, and story generation.**

**Rating:** 7
**Confidence:** 5
**Correctness:** Yes
**Clarity:** Yes

**Strengths:**

1. With the rapid development of the video generation field, previous metrics such as FVD are expected to become inadequate for evaluating future video models. Once the image quality of videos reaches a certain standard, the demand for video storytelling will become increasingly prominent. This article presents a forward-looking approach and proposes three reasonable metrics of increasing difficulty. Undoubtedly, it will have a significant impact on the advancement of the video generation field.
2. This paper proposes a set of universal algorithms for fine-grained annotation of existing video datasets. The automated annotation pipeline makes it easy for others to follow this work and create even larger and more comprehensive benchmarks.

**Additional Feedback:**

1. The article mentions the use of Google's PaLM in the data annotation pipeline, which has 540 billion parameters. Even if it is only used for inference, it can still be computationally expensive. However, the article does not compare whether other LLMs can achieve the same results.
2. The data annotation pipeline proposed in the article may still be effective for videos with complex characters, scenes, actions, and even camera jumps, such as those found on social media platforms or in movies. However, it may require additional modifications or enhancements to adapt to the increased complexity of such videos. Further research and experimentation will be needed to determine the pipeline's effectiveness in these scenarios.

**Documentation:**

Yes

**Limitations:**

The paper lacks an analysis of the robustness of the current automatic annotation pipeline, as well as a presentation and statistical analysis of any incorrectly labeled samples.

**Opportunities For Improvement:**

1. The number of videos in the benchmark is relatively small, with only about 6,000 videos in total across the three datasets, all of which are obtained from real-world footage. However, the current demand for video generation models is often to generate videos that are rarely or even never seen in the real world. Therefore, this benchmark may have some bias in evaluating the generation capabilities of models.
2. The scenes and characters in the videos are often relatively fixed, while videos on social media platforms or in movies often have a richer variety of scenes and changes in personnel, as well as camera jumps. Therefore, this dataset still has some distance to the evaluation metrics and standards of real industrial-level videos. Despite this, it is still an excellent piece of work.

**Relation To Prior Work:**

Yes

**Summary And Contributions:**

This paper presents a benchmark for story generation, consisting of three tasks of increasing difficulty: single-action generation, sequential-action generation, and story generation. The authors used the LLM to further annotate the original dataset, resulting in more specific labels for actions and timestamps. This allows for a more accurate evaluation of the effectiveness of the three tasks. After fine-tuning Phenaki using their annotated dataset, the authors believe that such a dataset can enable video generation models to attain superior video generation capabilities.

---

> ### Author Response · Authors · 2023-08-21
> **Rebuttal by Authors [1/2]**
>
> Thank you for thoroughly reviewing our work! We are happy to see you believe it will have a significant impact on the advancement of video generation, with a forward-looking approach with three tasks of increasing difficulty for video storytelling. We found your feedback for improvement accurate, and we addressed it as follows.
>
> > the current demand for video generation models is often to generate videos that are rarely or even never seen in the real world. Therefore, this benchmark may have some bias in evaluating the generation capabilities of models.
>
> This is true, yet we believe that both unseen and real-world video generation abilities are important. For instance, generating realistic transitions is one of the key challenges faced by these models. Moreover, we believe that real-world data is a great candidate to benchmark video generation models: we want to build systems that perform realistic actions, with entities that do not change throughout the video and where the background is consistent (e.g., if there’s wind). Real-world data allows us to benchmark all these aspects and also allows us (i) to make it easier for humans to evaluate, and (ii) to define less ambiguous instructions, which might be interpreted and visualized in different ways. Nevertheless, we expect that advancements as measured in StoryBench will likely transfer to more creative settings.
>
> Behind the scenes, we also wanted to include the Flintstones dataset in our benchmark, as a different type of visual data (i.e., cartoons) that would be closer to current interest in text-to-video models. However, when we reached out to the authors for more information on the license of the data, they replied that they “have been advised to not share the dataset anymore due to legal reasons. The links have been removed.”
>
> Having said that, we agree with the reviewer, and we have added this as an additional limitation of our work (text in purple in L350-352).
>
> > The scenes and characters in the videos are often relatively fixed, while videos on social media platforms or in movies often have a richer variety of scenes and changes in personnel, as well as camera jumps.
>
> We totally agree. We had stressed this among the Limitations of our work, and we have now added a reference in the Conclusion section (see text in purple in L372-374).
>
> > the article does not compare whether other LLMs can achieve the same results.
>
> Due to time limitations, we only relied on our best in-house model. However, given the tremendous progress that LLMs are experiencing, we believe that recent, smaller, instruction-tuned models would be able to perform as well on the task of sentence splitting. We release our LLM-generated captions, so that the community can (i) compare them against the ones generated with a different LLM, and (ii) simply replace them and run our full algorithmic pipeline with a single script.
>
> > data annotation pipeline proposed in the article may still be effective for videos [...] on social media platforms or in movies. However, it may require additional modifications or enhancements to adapt to the increased complexity of such videos. Further research and experimentation will be needed to determine the pipeline's effectiveness in these scenarios.
>
> We agree, and we had listed this as one of our Limitations. In this paper, we opted for careful collection of evaluation data as we believe that evaluation sets should be as rigorous and high-quality as possible, given the reliance on benchmarks to measure progress in AI research. We agree that it would be great to devise a more efficient pipeline should it result in data of similar quality. Moreover, one could experiment with off-the-shelf models that could facilitate data collection, especially for large-scale, noisier training data. We have now remarked this point in the Conclusion section (see text in purple in L372-374).

---

> ### Author Response · Authors · 2023-08-21
> **Rebuttal by Authors [2/2]**
>
> > The paper lacks an analysis of the robustness of the current automatic annotation pipeline
>
> Great point, thank you! We have added robustness analyses to the paper (App D.5).
> Comparing human annotations against the automated pipeline for Oops validation (1578 stories), we found that:
> - The distributions of the number of sentences per story of the two approaches are very similar (see Fig11 left). In particular, we notice that our method tends to split captions into 2 or 3 segments more often than the human annotators, who instead prefer not to split a sentence more often than our automatic pipeline.
> - The words corresponding to each video segment are very similar between human and automatic annotations. To assess this, we consider the subset of captions that have been split into the same number of sentences, so we can compute 1:1 mappings between the human and the algorithmic captions. Here, we observe a BLEU4 score of 63.6% (BLEU4 computes the similarity of 4-grams in the two captions), indicating a relatively high overlap of generated sentences to human references. We also note that humans were asked to enrich the original Oops and UVO captions with context information (e.g., what other relevant actors are doing), which our algorithmic pipeline does not explicitly tackle, leaving room for improvement in future work.
> - Finally, in Fig 11 right, we also see that the resulting duration of the algorithmically generated segments are slightly longer.
>
> > [The paper lacks] a presentation and statistical analysis of any incorrectly labeled samples
>
> We note that we re-use the same key-frame selection that was collected in VidLN (Voigtlaender et al., CVPR’23). As for our collected data – video descriptions and corresponding timestamps – we have measured high-quality metrics in data samples as follows.
>
> First, annotators were only moved to the final data annotation process after having completed a training stage successfully. Second, we asked annotator managers to verify the quality (i.e., the descriptions and timestamps match the narratives of the videos) of the final data by manually checking 25% of the data samples, where they found 97% success rate.
>
> We have added such information to App D.1 (see text in purple in L1038-1042).

---

### Official Review · Reviewer_V9cT · 2023-07-20
**a novel benchmark for text-to-video generation**

**Rating:** 7
**Confidence:** 3
**Correctness:** Yes.
**Clarity:** Yes.

**Strengths:**

This paper stands out for its contribution to the text-to-video generation research. The extensive, rich annotations and well-structured 'stories' is applicable in numerous sectors such as content creation, design, and art. The paper contains clear definition of tasks, evaluation methods, and presentation of results.

**Additional Feedback:**

please see above

**Documentation:**

Yes, all materials are nicely provided.

**Ethics:**

No.

**Limitations:**

Yes, the authors made it clear that the limitations of storybench include its reliance on short, user-generated videos, the scalability of its annotation framework, smaller baseline models, and discrepancies between automated metrics and human ratings.

**Opportunities For Improvement:**

This work can be further improved by expanding the dataset to include longer and more varied videos to enhance its applicability in realistic filmmaking applications. The annotation framework could also be made more efficient to accommodate larger and diverse datasets. Opportunities for enhancement also exist in training larger text-to-video models, aligning them with the scale of current text-to-image models.

**Relation To Prior Work:**

Yes.

**Summary And Contributions:**

This paper introduces a benchmark to stimulate text-to-video generation research. The authors enrich three existing video datasets with dense annotations, transforming them into time-stamped action sequences or "stories", and propose a method to convert grounded video captions into these story structures. The proposed dataset outlines three tasks of ascending difficulty for continuous story visualization. Evaluation is done through a trained Phenaki model in three different setups and a human study.

---

> ### Author Response · Authors · 2023-08-21
> **Rebuttal by Authors**
>
> Thank you for reviewing our work, we agree with your suggestions for improvement (see below) and are glad to see that you found our work outstanding, with extensive annotations, and tackling a variety of scenarios for text-to-video generation with clear tasks and evaluation methods.
>
> > This work can be further improved by expanding the dataset to include longer and more varied videos to enhance its applicability in realistic filmmaking applications.
>
> We totally agree. We had stressed this among the Limitations of our work, and we have now added a reference in the Conclusion section (see text in purple in L372-374).
>
> > The annotation framework could also be made more efficient to accommodate larger and diverse datasets.
>
> We agree, and we had listed this as one of our Limitations. In this paper, we opted for careful collection of evaluation data as we believe that evaluation sets should be as rigorous and high-quality as possible, given the reliance on benchmarks to measure progress in AI research. We agree that it would be great to devise a more efficient pipeline should this result in data of similar quality. Moreover, one could experiment with off-the-shelf models that could facilitate data collection, especially for large-scale, noisier training data. We have now remarked this point in the Conclusion section (see text in purple in L372-374).
>
> > Opportunities for enhancement also exist in training larger text-to-video models, aligning them with the scale of current text-to-image models.
>
> We couldn’t agree more: collecting large-scale, high-quality video–text data is a crucial step in improving the quality of text-to-video models. This is in line with our proposed “training data challenge,” which aims at fostering novel ideas to create good (story-like) text-to-video data.

---

### Official Review · Reviewer_xof8 · 2023-07-26
**Comments on StoryBench**

**Rating:** 7
**Confidence:** 5
**Clarity:** This paper is well written.

**Strengths:**

1. This paper targets an important issue, i.e., a benchmark for video generation, and proposes three typical video generation tasks for evaluation.
2. This paper prepares three datasets to evaluate the performance.


**Additional Feedback:**

Please refer to my weakness part.

**Correctness:**

Yes
A dataset and a evaluation method are prposed in a correct and a sound way.

**Documentation:**

Yes

**Ethics:**

 No ethical concerns

**Limitations:**

Yes. The authors have adequately addressed the limitations and potential negative societal impact of their work

**Opportunities For Improvement:**

1. The scale of the dataset is considerable given the high requirements of the data, but the quality of videos in benchmarks is limited, especially in terms of the resolution and image quality except for the limited analyzed by the authors. This can seriously limit the performance of evaluation.
2. As the authors take both 5 automatic evaluation metrics and user study evaluation, the analysis of some key points is missing, e.g., the meaning of each metric and their composition, how to explain the different results obtained from human and automatic algorithms, and which metric is more recommended for evaluating model performance.
3. There should be an analysis of data bias, since the creation of datasets is not an easy pipeline, and some operations can be noisy like key-frame selection and description.
4. This paper only adopts a transformer-based model in experiments, and I recommend adding some other popular generation baselines, like diffusion models.


**Relation To Prior Work:**

Yes

**Summary And Contributions:**

The paper introduces "StoryBench," a challenging benchmark for generating video stories from text prompts. It includes three tasks of increasing difficulty: action execution, story continuation, and story generation. The authors evaluate text-to-video models using the benchmark and highlight the benefits of training on algorithmically generated story-like data. They also emphasize the need for improved automatic metrics for video generation evaluation. StoryBench aims to foster future research in this exciting area of text-to-video generation.

---

> ### Author Response · Authors · 2023-08-21
> **Rebuttal by Authors**
>
> Thank you for carefully reviewing our paper! We are glad to see that our datasets and benchmark target an important issue in a sound way. You raised good points for improvement, which we have tackled as follows.
>
> > the quality of videos in benchmarks is limited, especially in terms of the resolution and image quality [...] This can seriously limit the performance of evaluation.
>
> We note that ground-truth videos have much higher resolution than the generated ones. In particular, for automatic evaluations, while we downsample them temporally to our model’s fps (8), we keep their original spatial resolutions. For instance, these are the most common resolutions for each Dev set:
> - Oops-CSV Dev: 448 videos at 1280x720, 105 videos at 404x720, 29 videos at 1276x720, 26 videos at 718x720, …
> - UVO-CSV Dev: 419 videos at 854x480, 285 videos at 640x480, 72 videos at 480x854, 35 videos at 852x480, …
> - DiDeMo-CSV Dev: 339 videos at 360x640, 259 videos at 480x640, 37 videos at 240x320, 14 videos at 362x640, …
>
> We have revised the paper to make this more clear (see text in purple on page 8).
>
> > the meaning of each metric and their composition
> We discussed human evaluation metrics in Sec 3.3, and automatic ones in Sec 3.4.
> To some degree, we expect that:
> - FID would measure “visual quality” since it compares the distribution of ground-truth frames with that of generated frames
> - FVD would measure “entity consistency” and “action realism” since it compares the distribution of ground-truth videos with that of generated videos
> - SIM would measure “visual quality”, “entity consistency”, “background consistency”, and “text adherence” as it compares ground-truth and generated frames 1:1
> - VTM would measure “text adherence” as it compares generated videos to their prompts
> - PQA would measure “visual quality” and “action realism” as it was trained to predict the average human subjective perception of a video.
>
> We have included this in App. D.4.
>
> > how to explain the different results obtained from human and automatic algorithms
>
> In Section 6.2, we note surprising differences between human and automatic ratings.
> As we discuss in our Limitations, this could be due to suboptimal automatic metrics. However, we also note that these neural models might give better signals if the generated videos were closer to the ones they were trained on (i.e., higher resolution and frame rates). As such, we cannot yet conclude why human and automatic metrics diverge. Rather, we encourage the community to report both kinds of metrics so that we can, collectively, gain a better understanding of this relationship.
>
> > which metric is more recommended for evaluating model performance
>
> At such an early stage of developing models and metrics for video generation, we encourage the use of human evaluation as a more reliable evaluation method. However, as written above, we encourage the community to also evaluate using automatic metrics in order to better understand and develop scalable solutions to assess generated videos. We have stressed this point in the Conclusion of the paper (see text in purple in L367-370).
>
> > analysis of data bias, since the creation of datasets is not an easy pipeline
>
> We remark that our main contributions in data collection are related to the descriptions of the videos, their division into single-action segments and the labeling for diagnostics. We discuss the issues with biases in data and models in the Ethics section (App. B). Nevertheless, we agree with you that an in-depth data bias analysis of the videos in StoryBench would be an important contribution in future endeavors.
>
> > some operations can be noisy like key-frame selection and description
>
> We note that we re-use the same key-frame selection that was collected in VidLN (Voigtlaender et al., CVPR’23). As for our collected data – video descriptions and corresponding timestamps – we have measured high-quality metrics in data samples as follows.
>
> First, annotators were only moved to the final data annotation process after having completed a training stage successfully. Second, we asked annotator managers to verify the quality (i.e., the descriptions and timestamps match the narratives of the videos) of the final data by manually checking 25% of the data samples, where they found 97% success rate.
>
> We have added such information to App. D.1 (see text in purple in L1038-1042).
>
> > This paper only adopts a transformer-based model in experiments, and I recommend adding some other popular generation baselines, like diffusion models.
>
> Unfortunately, beyond Phenaki, we are not aware of any existing model that is capable of dynamically extending videos given a sequence of prompts and their duration. Current (diffusion) models can only generate a fixed-length video, and they are not open-source. The goal of our benchmark is indeed to stimulate research on creating such generative video models for real-world use cases and that can better be controlled by their users.

---

> > ### Comment · Reviewer_xof8 · 2023-08-28
> >
> > The authors have addressed some of my concerns in the rebuttal. I will keep my rating of acceptance.

---

### Official Review · Reviewer_Ae3X · 2023-08-03
**Comments on StoryBench**

**Rating:** 6
**Confidence:** 4
**Correctness:** Yes
**Clarity:** Yes

**Strengths:**

1. This work is well presented and well-written.
2. Automated metrics in video generation from text is important.
3. The benchmark model is well-trained with enough video data.

**Additional Feedback:**

NA

**Documentation:**

Yes

**Limitations:**

Yes

**Opportunities For Improvement:**

The proposed PHENAKI-GEN adopt caption-video pairs as training data. My main concern is that the caption may not good prompts for video generation.

**Relation To Prior Work:**

Yes

**Summary And Contributions:**

This study presents StoryBench, a multifaceted benchmark for continuous story visualization aimed at evaluating the performance of text-to-video models. By collecting a human-annotated dataset and introducing three video generation tasks, including action execution, story continuation, and story generation, the authors evaluate small but powerful text-to-video baseline models and propose guidelines for human evaluation and improved video generation.

---

### Decision · Program_Chairs · 2023-09-22

**Decision:**

Accept (Poster)

**Comment:**

All reviewers agreed that a benchmark for video generation is an important problem and the dataset gathered is good, proposing three video generation tasks and datasets for evaluation.
The paper is well written and clear.
All reviewers were all in favor of accepting the paper.
Many of the reviewers' concerns were nicely addressed in the rebuttal and the authors promised to add/correct them in the final version and clarify what needs to be clarified.
Therefore, this paper is recommended for acceptance - provided that the authors follow in the final revision the improvement suggestions by reviewers and in their rebuttal.